# Estimating disease prevalence in large datasets using genetic risk scores

Benjamin D. Evans [1,2,3,8], Piotr Słowiński [1,4,8], Andrew T. Hattersley [5,6], Samuel E. Jones [5], Seth Sharp[5], Robert A. Kimmitt[5,6], Michael N. Weedon[5], Richard A. Oram [5,6], Krasimira Tsaneva-Atanasova [1,7] & Nicholas J. Thomas [1,2,6✉]

Clinical classification is essential for estimating disease prevalence but is difficult, often requiring complex investigations. The widespread availability of population level genetic data makes novel genetic stratification techniques a highly attractive alternative. We propose a generalizable mathematical framework for determining disease prevalence within a cohort using genetic risk scores. We compare and evaluate methods based on the means of genetic risk scores' distributions; the Earth Mover's Distance between distributions; a linear combination of kernel density estimates of distributions; and an Excess method. We demonstrate the performance of genetic stratification to produce robust prevalence estimates. Specifically, we show that robust estimates of prevalence are still possible even with rarer diseases, smaller cohort sizes and less discriminative genetic risk scores, highlighting the general utility of these approaches. Genetic stratification techniques offer exciting new research tools, enabling unbiased insights into disease prevalence and clinical characteristics unhampered by clinical classification criteria.

[1] Department of Mathematics, University of Exeter, North Park Road, Exeter EX4 4QF, UK. [2] Living Systems Institute, Centre for Biomedical Modelling and Analysis, University of Exeter, Stocker Road, Exeter EX4 4QD, UK. [3] School of Psychological Science, University of Bristol, Priory Road, Bristol BS8 1TU, UK. [4] Living Systems Institute, Translational Research Exchange @ Exeter, University of Exeter, Stocker Road, Exeter EX4 4QD, UK. [5] University of Exeter Medical School, Institute of Biomedical & Clinical Science, RILD Building, Royal Devon & Exeter Hospital, Barrack Road, Exeter EX2 5DW, UK. [6] Royal Devon & Exeter NHS Foundation Trust, Exeter, UK. [7] Living Systems Institute, EPSRC Hub for Quantitative Modelling in Healthcare, University of Exeter, Stocker Road, Exeter EX4 4QD, UK. [8] These authors contributed equally: Benjamin D. Evans, Piotr Słowiński. ✉email: n.thomas3@exeter.ac.uk

The development and refinement of polygenic analysis techniques is greatly increasing our understanding of many diseases. Using polygenic risk has allowed insights into disease etiology and through Mendelian randomization evaluation of causality[1]. Clinically, capturing polygenic susceptibility through genetic risk scores (GRS) can be used to determine individuals at the highest risk of a disease[2–4]. This paper concentrates on an innovative use of polygenic risk to genetically estimate disease prevalence (proportion of individuals with and without a disease) within a cohort. Currently estimating disease prevalence is difficult as it requires robust clinical classification. Disease-specific investigations are rarely available in population-level data and inaccuracies associated with self-reported diagnosis are well recognized[5,6]. Given the increasing availability of population-level genetic data, novel polygenic estimates of disease prevalence are an extremely attractive alternative.

The basis of genetically determining disease prevalence is fundamentally that the distribution of a specific disease GRS within a cohort will reflect the mixture of GRS of those with the disease (cases) and those without (non-cases). This mixture GRS distribution will lie between reference groups of cases and non-cases and will reflect the relative proportion of cases to non-cases (Fig. 1a). The location of the mixture cohort's GRS distribution in comparison to the GRS distribution of known cases and non-cases allows the respective proportion of each group to be determined which provides a genetic-based estimate of disease prevalence. Furthermore, using the genetically calculated proportion of a disease within a cohort allows the additional benefit of associated clinical features of the genetically defined disease cohort to be determined. It is worth emphasising that in almost all polygenic risk situations, even those at the highest genetic risk are unlikely to develop the relevant disease and therefore this concept does not remain valid at an individual level. Nonetheless, at a group level the average GRS will be higher in a cohort with disease versus those without.

In this paper we assess, the performance and utility of polygenic stratification as a tool for determining disease prevalence. Through simulated scenarios and real-world data, we evaluate different mathematical techniques for determining disease prevalence based on the GRS distribution within a cohort. The generalizability and robustness of genetic stratification have been investigated through a systematic evaluation of the cohort characteristics required for estimates to remain robust. Specifically, the impact on the performance of the prevalence of the disease, the mixture cohort size and the strength of genetic predisposition for a disease. Finally, in order to highlight the utility of the proposed framework we apply our methodologies in the context of identifying the prevalence of undiagnosed coeliac disease within a cohort adhering to a gluten-free diet.

**Genetic stratification summary**. We present three methods developed to estimate the proportions of cases and non-cases in an unknown mixture cohort using GRS distributions and compare them with the published Excess approach[7]. The methods' performance characteristics are evaluated over clinically relevant parameter ranges using GRS for type 1 diabetes (T1D), type 2 diabetes (T2D) and coeliac disease, as well as synthetic data. Clinical sample sets were taken from the following cohorts: T1D ($n = 1,963$) and T2D ($n = 1,924$) from the Wellcome Trust Case Control Consortium (WTCCCC)[8], Coeliac disease reference cases ($n = 12,018$) from a combination of European studies[9] with non-cases (controls) and mixture (gluten-free diet) cohorts from UK Biobank ($n = 12,000$ and $n = 12,757$, respectively)[10].

To compare the methods under different conditions, the T1DGRS data were split in half to form reference cohorts and an

independent hold-out set for generating parameterised mixture cohorts (Figs. 2 and 3). In these analyses, mixtures were constructed by sampling with replacement, enabling larger mixture sizes to be used than the size of the hold-out sets from which they were derived[11,12]. Synthetic data sets were also constructed from Gaussian distributions of equal standard deviations (set to 1) but different means (see Table 1 and Fig. 4). For the reference cohort of cases, $R_C$, the mean of the generating distribution was always 0 while the mean for the non-cases cohort, $R_N$, was systematically varied in order to investigate the effect of differences in discriminability signified by the area under the curve (AUC) of the GRS distribution. For further details of the data sets, see "Methods".

In each method, two cohorts consisting of the GRS of individuals with and without a particular polygenic disease were taken as references, denoted $R_C$ (the reference cohort of cases) and $R_N$ (the reference cohort of non-cases). The proportions of individuals from these reference cohorts (denoted $p_C$ and $p_N$ respectively) who comprise an unknown mixture cohort ($\widetilde{M}$) were estimated based on the properties of the reference cohorts. When only one proportion is mentioned, this is $p_C$ (i.e. relative to the reference cohort of cases, $R_C$), unless otherwise stated. The cohort characteristics used are dependent upon the particular method employed as illustrated in Fig. 1 and are detailed below.

Throughout this paper, we assume that the unknown mixture cohort is composed solely of the samples that come from the two reference cohorts (blue and red dots in Fig. 1). In practice, this means that $p_N$ (prevalence of non-cases) and $p_C$ (prevalence of cases) sum to one, $p_N + p_C = 1$, and so accordingly, the proportion of non-cases was calculated as: $p_N = 1 - p_C$. Furthermore, the presented Earth Mover's Distance (EMD) and Kernel Density Estimation (KDE) methods make it possible to check if this assumption is satisfied. We revisit details of such checks in the discussion and supplementary information.

Finally, our methods are all based on the assumption that between the reference and mixture cohorts, cases and non-cases are genetically equivalent. This assumption must hold true for estimates to be valid and becomes less certain if the mixture cohort is derived from a different population than those used for reference. For this reason, we recommend these methods should be used to estimate disease prevalence within a subset of a population where reference cases and non-cases can be derived from the same population, for example, UK Biobank. This does not completely exclude using reference cohorts derived from different datasets, particularly where robust disease cases may be difficult to define[7], but in this context, extreme caution should be exercised prior to applying the methods and around the interpretation of the generated estimates. Using reference cohorts from a different population from the mixture analysis should only be undertaken following close examination of the selection criteria and demographics of the reference and mixture cohorts to ensure equivalence. This is of particular importance when studying different geographical populations where allele frequencies are known to vary[13,14]. Accordingly, in this manuscript all analyses are restricted to white Europeans; the populations that the reference GRS distributions were derived from. Where possible, the GRS of the reference non-cases (controls) and cases should be compared with the GRS of known non-cases and cases within the same population the mixture has been taken from. This could be done, for example, by means of a statistical test appropriate for the assessment of the observed GRS distributions. An example of the importance of this and how it can be detected is demonstrated by the T2DGRS for a reference T2D population from the WTCCC[8]. The WTCCC cohort was largely selected based on a positive family history of T2D or early disease onset

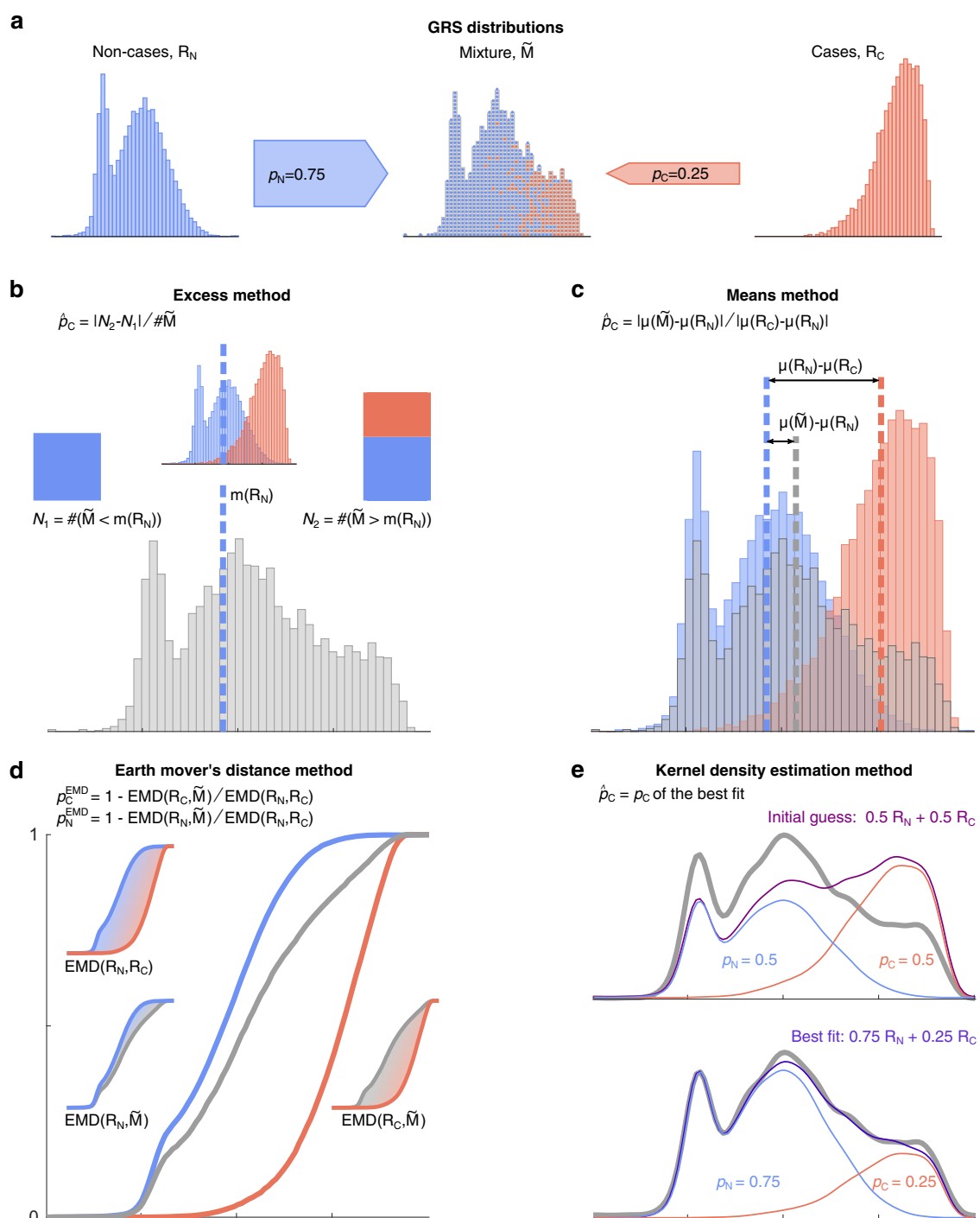

**Fig. 1 Illustration of a mixture population drawn from two reference populations and the four proportion estimation methods. a** This mixture population emulates the real-world scenario of a population composed solely of individuals drawn from each subpopulation of non-cases ($R_N$, blue) and cases ($R_C$, red). Mixture ($\tilde{M}$) cohort possesses features of both reference cohorts. Each method uses different characteristics of the mixture and reference cohorts to estimate the proportion of constituents of the mixture cohort ($p_C$ and $p_N$). The Excess method (**b**) considers the number of cases above the mixture median in excess of those expected in a pure control (non-cases) reference cohort. The Means method (**c**) uses the normalised difference of the mixture cohort's mean and the two reference cohorts' means. The Earth Mover's Distance method (**d**) uses the weighted costs of transforming the mixture distribution into the reference distributions. The Kernel Density Estimation method (**e**) fits smoothed templates to each of the reference distributions and then fits a weighted sum of these templates to the mixture distribution, adjusting the amplitudes of each with the Levenberg–Marquardt algorithm. Figure generated using artificial data.

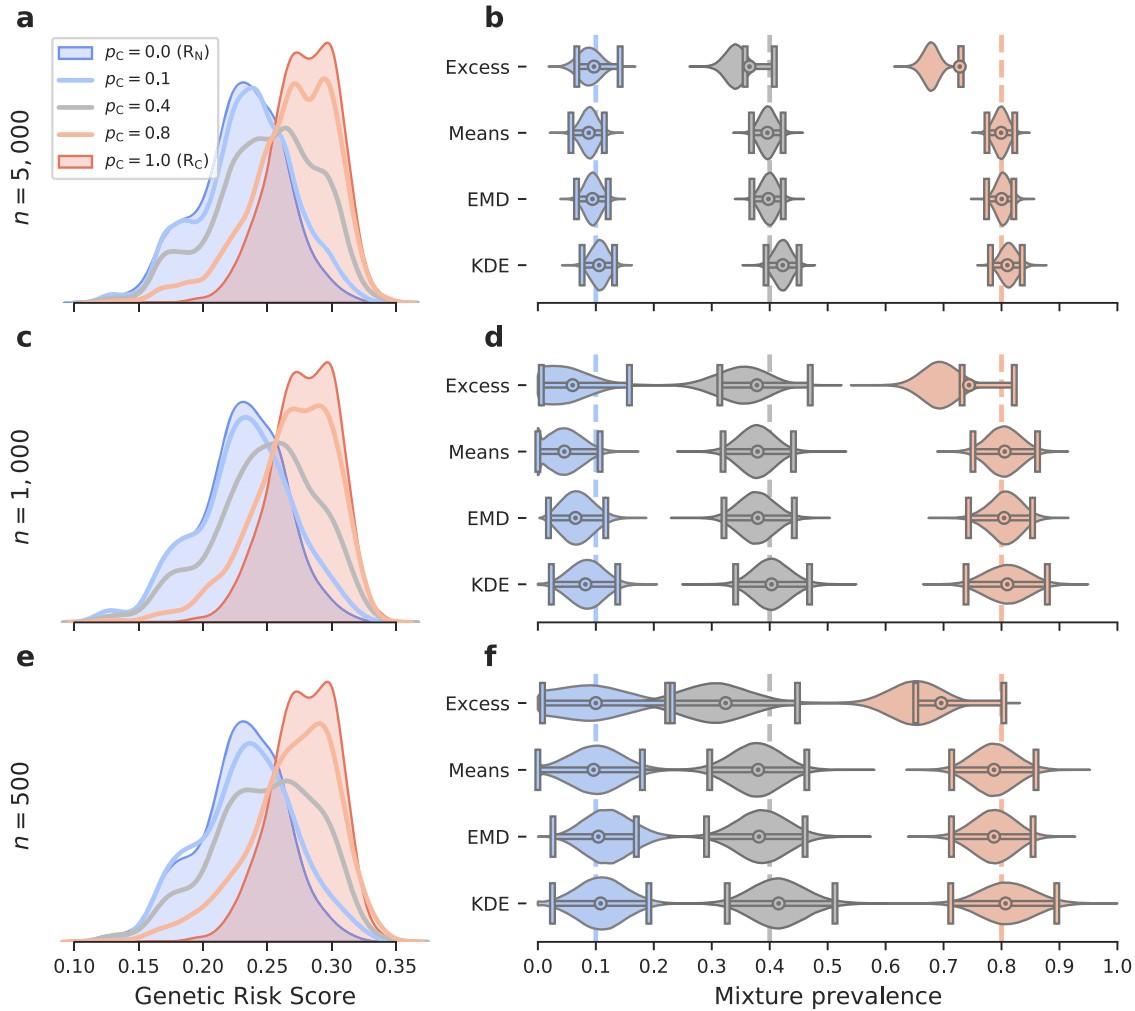

**Fig. 2 A comparison of the four methods prevalence estimates and confidence intervals for varying proportion of cases and for three sample sizes.** Mixture distributions of non-cases and T1D patients from WTCCC[8] were constructed with $p_C = \{0.1, 0.4, 0.8\}$ (shown in blue, grey and red respectively) and $n = 500, 1000, 5000$ (shown in panels (**e–f**), (**c–d**), (**a–b**) respectively). **a, c, e** The constructed mixture distributions and reference distributions ($R_C$, shaded red and $R_N$, shaded blue) from which they were constructed. **b, d, f** Prevalence estimates, $\hat{p}_C$ (bullseye) obtained by each of the four methods for varying $p_C$ (x-axis) and cohort size, $n$ (rows). Each estimated $\hat{p}_C$ value is shown together with a violin plot illustrating the distribution of the 100,000 estimates of prevalence ($p'_C$) in the bootstrap samples and with confidence intervals ($\alpha = 0.05$) shown as horizontal lines with vertical bars at the ends. Dashed vertical lines indicate reference prevalence values $p_C$. In all the cases, for the Excess method we observe a large offset between the violin plots (including confidence intervals) and the $\hat{p}_C$ value. This offset is a result of the systematic bias of the Excess method. The other three methods generally show much less bias. Sample sizes: $R_C$ – cases WTCCC T1D ($n = 982$), $R_N$ – non-cases WTCCC T2D ($n = 962$), mixtures – sampled with replacement from a holdout half of the $R_C$ ($n = 981$) and $R_N$ ($n = 962$) samples.

and is therefore enriched for T2D risk variants. As shown in Supplementary Fig. 1 the distribution of T2DGRS of unselected T2D cases from population data in UK Biobank is significantly lower than the T2DGRS in the WTCCC T2D reference. The T2DGRS in UK Biobank population T2D cases only becomes equivalent to the WTCCC when the same case selection criteria are mirrored. If this WTCCC cohort was used as a reference T2D population when evaluating the prevalence of T2D in a cohort in the UK biobank, it would have influenced the accuracy of estimates since it does not constitute a representative T2D cohort.

**The Excess method**. This estimates the proportion from the number of excess disease cases above the mixture cohort's median score compared to the equal numbers expected in a pure control cohort (Fig. 1b). We illustrate the method as introduced in ref. [7].

**The Means method**. This compares the mean GRS of the mixture cohort to the means of the two reference cohorts and estimates the mixture proportion according to the normalised difference between the two (Fig. 1c).

**The Earth Mover's Distance (EMD) method**. This uses the weighted cost of transforming the mixture distribution into each reference distribution (more formally, the integral of the difference between the cumulative density functions, i.e., the area between the curves). This method allows $p_N$ and $p_C$ to be computed independently (Fig. 1d) and so provides a way to validate the assumption that the mixture is composed solely of the samples from the two reference cohorts, $\hat{p}_N + \hat{p}_C = 1$; if the sum is significantly different from 1, then the assumption is not satisfied. In this study, we use the mean of the two estimates for $p_C^{EMD}$ and $1 - p_N^{EMD}$ as the estimate of the $\hat{p}_C$.

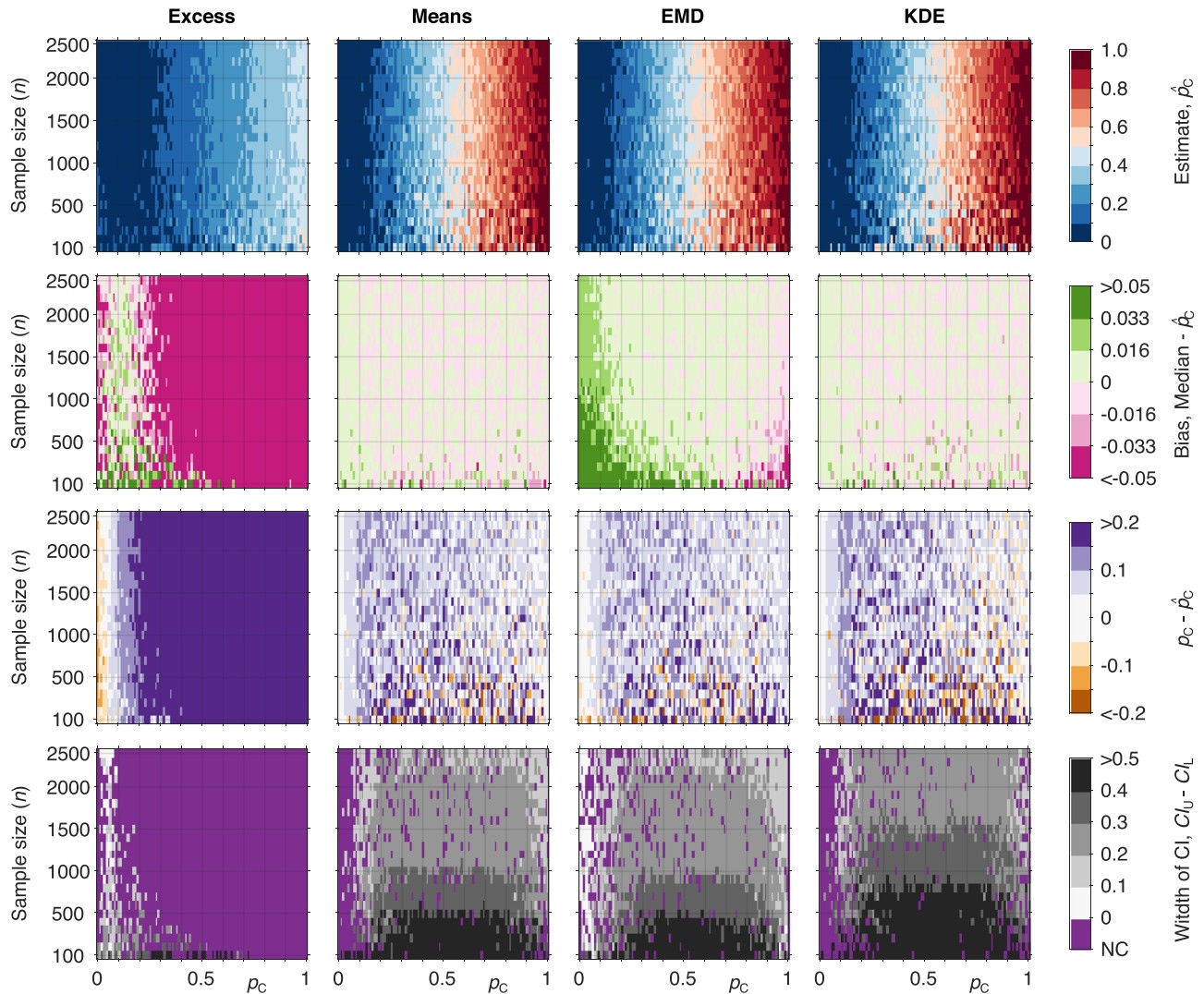

**Fig. 3 A comparison of the four methods with prevalence estimates and confidence intervals for varying proportion of disease and cohort sizes using the (Type 2 GRS) from the WTCCC dataset: T1D($n = 1,963$), T2D($n = 1,924$).** (Top row) Estimate of prevalence ($\hat{p}_C$) in the constructed mixtures. (Second row) Bias of the prevalence estimates ($\hat{p}_C$) across the constructed mixtures. (Third row) Deviation from the true proportion ($p_C - \hat{p}_C$) across the constructed mixtures. (Bottom row) The width of confidence intervals ($CI_U - CI_L$) of the estimates across the constructed mixtures. The purple colour (bottom row) indicates regions in which the confidence interval did not include the true value ($p_C$), $CI_L = CI_U$ or the confidence interval was undefined (both latter cases can happen if $\hat{p}_C = 0$ or $\hat{p}_C = 1$). Sample sizes: $R_C$ - cases WTCCC T2D ($n = 962$), $R_N$ - non-cases WTCCC T1D ($n = 982$), mixtures – sampled with replacement from a hold-out half of the $R_C$ ($n = 962$) and $R_N$ ($n = 981$) samples; see "Methods" for details.

**The Kernel Density Estimation (KDE) method.** This method fits a smoothed template to each reference distribution (by convolving each sample with a Gaussian kernel) and builds a model of the mixture as a weighted sum of these two templates. The method then adjusts the proportion of these templates with the Levenberg–Marquardt (damped least squares) algorithm until the sum optimally fits the mixture distribution (Fig. 1e), noting that the algorithm could find one of the potentially several local minima. In other words, the method finds (one of) the linear (convex) combination(s) of the reference distributions that best fits the mixture distribution.

## Results

### Performance of genetic stratification.
We start by using the T1D GRS (AUC = 0.88[2]) to evaluate the performance of all four methods on artificially constructed (synthetic) mixtures. The mixtures are generated by sampling with replacement from half of the reference data (holdout subset), to ensure the reference and

mixture cohorts are independent and identically distributed (for details see "Methods"). Figure 2 demonstrates that genetic stratification allows robust estimates of disease prevalence (proportion of cases to non-cases) around known values. The accuracy (defined as deviation from the true proportion) and precision (defined as confidence interval width) of estimates are dependent on the following variables: proportion of cases and non-cases within the mixture, the mixture size and the discriminative ability of the GRS. For each method, we describe how each of these variables affects the accuracy and/or precision of prevalence estimates.

**What is impact of the proportion of cases to controls in the mixture cohort?** In all methods except the Excess, away from extremes of proportion, varying the proportion of cases to controls has no impact on the accuracy or precision of prevalence estimates (Fig. 2). Using heat maps we illustrate the combined effect of gradually changing both proportion and mixture size on

**Table 1 The minimum mixture size required to give precision of ±0.05 around a prevalence of 0.1 with increasing AUC.**

| Method/AUC | 0.6 | 0.65 | 0.7 | 0.75 | 0.8 | 0.85 | 0.9 |
|---|---|---|---|---|---|---|---|
| Excess | – | – | – | – | 3200 | 3100 | 3100 |
| | Q25: – | Q25: – | Q25: – | Q25: – | Q25:3000 | Q25:2900 | Q25:3000 |
| | Q75: – | Q75: – | Q75: – | Q75: – | Q75:3400 | Q75:3300 | Q75:3200 |
| | 87/100 | 62/100 | 42/100 | 19/100 | 13/100 | 2/100 | 3/100 |
| Means | 26,500 | 11,500 | 6200 | 3700 | 2500 | 1700 | 1100 |
| | Q25:25,850 | Q25:11,100 | Q25:6000 | Q25:3600 | Q25:2400 | Q25:1600 | Q25:1100 |
| | Q75:27,300 | Q75:11,700 | Q75:6375 | Q75:3900 | Q75:2525 | Q75:1700 | Q75:1200 |
| | 4/100 | 1/100 | 1/100 | 0/100 | 3/100 | 4/100 | 1/100 |
| EMD | 25,500 | 10,800 | 5700 | 3400 | 2200 | 1500 | 1000 |
| | Q25:24,600 | Q25:10,400 | Q25:5550 | Q25:3300 | Q25:2100 | Q25:1400 | Q25:1000 |
| | Q75:26,300 | Q75:11,200 | Q75:6000 | Q75:3600 | Q75:2300 | Q75:1500 | Q75:1000 |
| | 0/100 | 0/100 | 0/100 | 1/100 | 0/100 | 2/100 | 0/100 |
| KDE | 38,250 | 17,000 | 9000 | 5500 | 3400 | 2100 | 1300 |
| | Q25:37,000 | Q25:16,000 | Q25:8500 | Q25:5300 | Q25:3300 | Q25:2100 | Q25:1300 |
| | Q75:39,500 | Q75:18,000 | Q75:9125 | Q75:5700 | Q75:3500 | Q75:2200 | Q75:1400 |
| | 54/100 | 17/100 | 15/100 | 4/100 | 3/100 | 0/100 | 0/100 |

The table shows the median minimum mixture size, 25% quantile, 75% quantile, and the number of misses (coverage probability)—when the confidence interval at the minimum mixture size did not contain the true prevalence value ($p_C = 0.1$); the minimum mixture size is based on 100 estimation runs (see Methods—Varying mixture size). The estimates based on the Excess method do not converge to $p_C = 0.1$ with increasing sample size for AUC = {0.6, 0.65, 0.7, 0.75}. The number of misses quickly increases to 100, showing that the estimates converge to a value much smaller than 0.1. The estimates based on the KDE method converge to $p_C = 0.1$ for AUC = 0.6. For further details of the computations see "Methods".

the accuracy of estimates (Fig. 3 and Supplementary Fig. 2). At extremes of proportion accuracy significantly reduces, tending to underestimate at high proportions and overestimate at low proportions. Increasing sample size reduces the extent to which proportions are classed as extremes thereby improving accuracy and precision for estimating the prevalence of rarer diseases. This is demonstrated by Fig. 2, a mixture size of 500 gives imprecise estimates around a 10% disease prevalence (proportion 0.1) and includes zero. Increasing the mixture size to 5,000 significantly improves the precision around the same 10% prevalence allowing a meaningful estimate of disease prevalence.

**What is the impact of the size of the mixture cohort?** With all but the smallest cohort sizes, prevalence estimates remain valid. Not surprisingly, increasing cohort size leads to an improvement in the precision of estimates, (Figs. 2 and 3). Increasing mixture size improves precision because larger cohort sizes can be seen to represent the characteristics of the reference distributions more accurately. Where larger mixtures cohort sizes are not possible, Fig. 3 clearly demonstrates that for all methods except the Excess, accurate albeit less precise, estimates of disease prevalence can still be achieved with lower case numbers. Figure 2 shows that using a T1DGRS and a mixture of just 500 cases can still provide accurate and clinically informative estimates around a disease prevalence of 40%, e.g., determining the prevalence of T1D in diabetes cases rapidly requiring insulin (clinical PPV of ≈50% for identifying T1D[15]).

**How predictive does a GRS need to be?** Accuracy and precision of estimates for all four methods reduce when using less discriminative GRS. However, excluding the Excess method, robust estimates of proportion are possible even when using GRS with AUC around 0.6 or above. This is demonstrated in Fig. 4 where we create artificial GRS with the area under the ROC curve (AUC) varying from completely non-discriminative (AUC = 0.5) to fully discriminative (AUC = 1). Reducing GRS AUC leads to widening of confidence intervals around evaluated disease prevalence's of 10% and 25%. The reduction in precision can be entirely mitigated by increasing the mixture cohort size. This is emphasised by Table 1, which shows the minimum mixture size required to give an estimated precision of 0.1 ($CI_U - CI_L$) around a prevalence of 0.1 with increasing AUC. For instance, using the

EMD method a mixture size of 25,500 (2,550 cases and 22,950 non cases) and an AUC of 0.6 allows robust precision around a 10% disease prevalence. A real-world clinical example is shown in Supplementary Fig. 5 accurately estimating the proportion of T2D cases in participants with self-reported glaucoma in UK Biobank using a less discriminative GRS (T2DGRS AUC 0.65, calculated in this study).

**What is the relative performance of the different methods?** We find that the Means, KDE and EMD methods perform well in estimating prevalence. Their accuracy and precision are largely comparable and all outperform the Excess method. The Excess method demonstrates reduced performance and exhibits strong bias (difference between the estimated prevalence $\hat{p}_C$ and the median of the bootstrap values $p'_C$) typically underestimating the true prevalence. Figure 3 shows that regardless of the mixture size, the Excess method is practically unusable for any but the highest AUC. The relatively comparable performance of the Excess method in Fig. 2 is a consequence of the high AUC of the T1DGRS (0.88[2]) and the strong asymmetry of the reference distributions.

**Clinical example estimating prevalence of coeliac disease.** Finally, we illustrate a worked example asking the question of how much-undiagnosed coeliac disease is present within a population adhering to a gluten-free diet (Fig. 5) using a coeliac disease GRS (CDGRS). This is important as whilst people observe a gluten-free diet for a number of reasons, it is possible that without getting a formal diagnosis people with undiagnosed coeliac disease eliminated gluten from their diet using trial and error to alleviate abdominal symptoms. For each method we: (1) compute an estimate of prevalence (2) use modelled mixtures and bootstrapping to calculate its confidence intervals. All methodologies provide estimates of the proportion of individuals with coeliac disease with their 95% CIs shown in square brackets: Excess = 15.0% [13.4%, 17.7%]; Means = 15.1% [13.5%, 16.6%]; EMD = 15.1% [13.5%, 16.7%]; KDE = 13.2% [11.6%, 14.7%]. In this same population in the UK biobank[10] adhering to a gluten-free diet, 13.9% of individuals were known coeliac cases (self-reported or ICD10 code; see "Methods" for details). Our results suggest an absence of undiagnosed coeliac disease in all patients

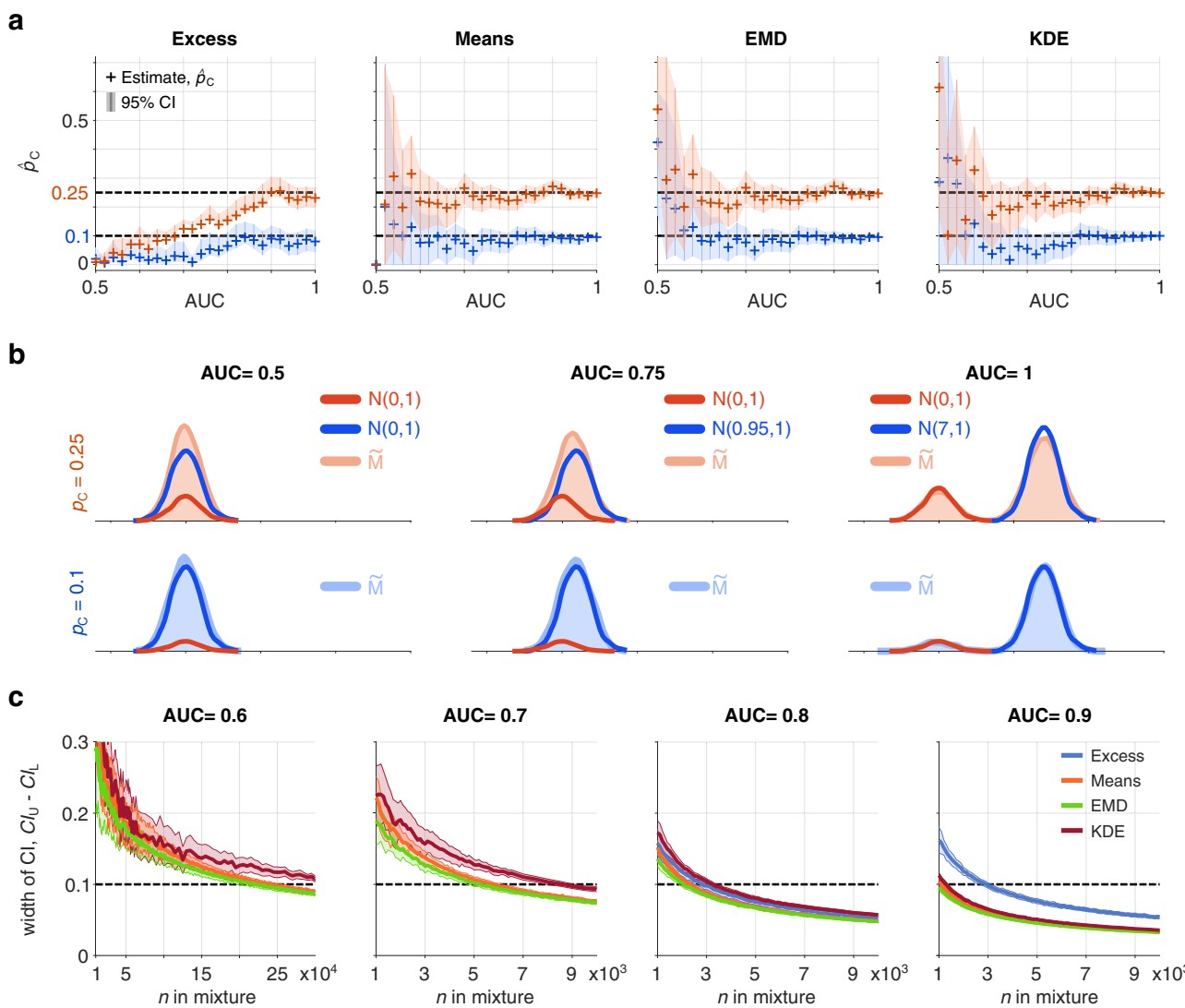

**Fig. 4 A comparison of the four methods using an artificial genetic risk score with increasing discriminative ability as measured by AUC, from AUC = 0.5 (no discriminative ability) through to AUC = 1, (complete differentiation). a** The estimated proportion (+ marker) with confidence intervals (vertical lines with shading) around $p_C = 0.1$ (blue) or $p_C = 0.25$ (red) for each of the methods (Excess, Means, EMD, KDE) are shown using mixture size, $n = 5,000$. **b** The constructed mixture distributions and reference distributions ($R_C$, shaded red and $R_N$, shaded blue) from which they were constructed for AUC = {0.5, 0.75, 1}. **c** Dependence of the width of CI ($CI_U - CI_L$) on the number of points in the mixture sample for AUC = {0.6, 0.7, 0.8, 0.9} and $p_C = 0.1$. Curves and shading show median ± standard deviation of the width of CI. The plot for the Excess method for AUC = {0.6, 0.7} is omitted because the method does not converge to $p_C = 0.1$. This figure is generated using artificial data: N($\mu,\sigma$) is a normal distribution with mean $\mu$ = {0.0, 0.08, 0.15, 0.22, 0.29, 0.37, 0.44, 0.51, 0.59, 0.66, 0.74, 0.82, 0.91, 0.99, 1.09, 1.19, 1.29, 1.4, 1.52, 1.65, 1.81. 1.98, 2.19, 2.47, 2.91, 7} and standard deviation $\sigma = 1$ and $\widetilde{M}$ is a mixture of the two normal distributions ($R_C$ is always N(0,1)). Both reference samples have $n = 2,000$. For AUC=0.5, means of the constructed mixture samples (for $p_C = 0.1$ and $p_C = 0.25$) were smaller than both means of the reference samples, in these cases the prevalence estimate from the Means method is assumed to be $\hat{p}_C = 0$ and confidence intervals are undefined due to undetermined acceleration value.

adhering to a gluten-free diet and not known to have the condition.

## Discussion

We present analysis of a novel approach to disease classification based on genetic predisposition. We demonstrate genetic stratification produces robust prevalence estimates even in the context of: rarer diseases, smaller cohort sizes and less discriminative GRS, highlighting the general utility of the proposed approaches. This was demonstrated through head-to-head evaluation of four methods including the original Excess methodology published by Thomas et al.[7]. The presented examples illustrate the performance and utility of these method across a range of different scenarios highlighting improved accuracy of the new approaches

over the original Excess method. We supplemented the estimation methods by combining Monte Carlo[11] sampling and bootstrap[12] methods to quantify uncertainty around the estimate and compute realistic confidence intervals.

**Distribution of GRS can be used to estimate disease prevalence within a cohort.** Our results show that robust estimates of prevalence are possible using differences in distributions of GRS between cohorts of cases and non-cases. Our methods build on the previously published genetic stratification by Thomas et al.[7]. This novel concept is important, as when coupled to the everincreasing availability of population-level genetic datasets, it allows fresh insights into disease epidemiology without requiring extensive investigations or unreliable self-reported diagnosis[5,6].

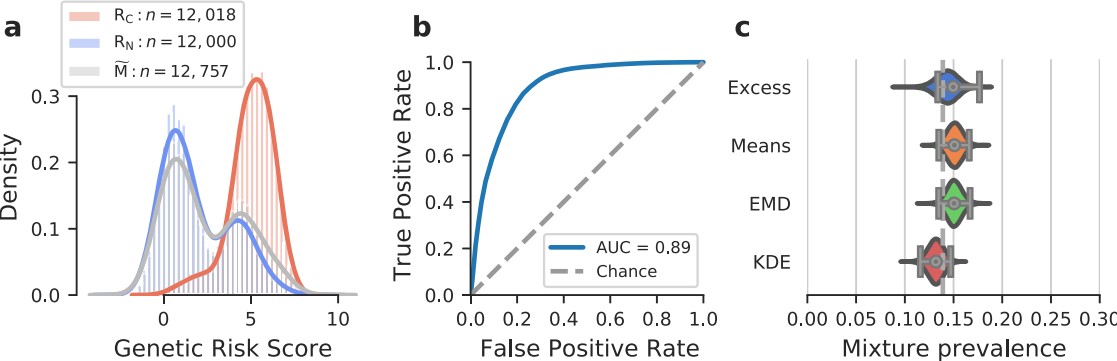

**Fig. 5 Coeliac disease dataset worked example.** A comparison of the four methods applied to a gluten-free cohort from the UK biobank (mixture population $\widetilde{M}$). **a** The reference and the mixture distributions ($R_C$, shaded red, $R_N$, shaded blue, $\widetilde{M}$ shaded grey, respectively). **b** A receiver operating characteristic (ROC) curve for the two reference distributions (blue). **c** Estimated values of prevalence $\hat{p}_C$ (grey bullseye) and 95% confidence intervals (horizontal lines with vertical bars at the ends) are plotted on the right showing estimates of Excess = 15.0%, Means = 15.1%, EMD = 15.1%, KDE = 13.2%. The violin plots show the distribution of the 100,000 estimates of prevalence ($p'_C$) in the bootstrap cohorts. The proportion of participants adhering to a gluten-free diet and reporting coeliac disease is shown as a dashed vertical line. Sample sizes: non-cases UK Biobank ($n = 12,000$), cases coeliac disease reference cohort ($n = 12,018$), mixture self-reported gluten-free diet UK Biobank ($n = 12,757$).

The permanence associated with genetic risk makes these methods potentially very powerful tools for clinical researchers and enables accurate evaluation where cases are difficult to differentiate clinically.

**Rare diseases and small mixture cohorts can be evaluated.** Accurate estimates were possible with mixture cohorts containing as few as 500 individuals and away from extremes of proportion disease prevalence had little impact. Precision around estimates improved with increasing cohort size. Larger mixture cohorts, readily achievable in modern-day population datasets (UK Biobank has genotyped ≈500,000 individuals[10]), almost entirely mitigated for the reduced precision observed with using less discriminative GRS (lower AUC). When disease prevalence is extremely low, robust estimates can still be achieved through mixture enrichment. This enrichment will inevitably be to the detriment of smaller mixture sizes but because proportions are moved away from extremes, in this situation accuracy is still improved.

**Estimates remain robust in diseases with less discriminative GRS.** Whilst accuracy and precision are higher when utilising a more discriminative GRS, we show that clinically meaningful estimates can still be obtained using GRS with AUC as low as 0.6. While in theory our methods could be used in diseases with minimal genetic predisposition (AUC < 0.6) our analysis would suggest extreme caution in these scenarios and that extremely large mixture sizes would be required to generate any clinically meaningful confidence around estimates. The performance of the Means, EMD and KDE methods is very good in the case of normal GRS distributions with equal standard deviations e.g., diseases with polygenetic risk arising from a large number of causal variants, each with tiny effects, e.g., T2D. In diseases where certain variants predominate, e.g., HLA in autoimmune disease, the GRS will be skewed to account for this, e.g., T1D. In this instance, the EMD and KDE methods will be more accurate, as they are able to utilise the unequal skewness (or other properties such as standard deviations or kurtosis) even when the means of the reference distributions are close, see Supplementary Fig. 7. In diseases where one variant has the predominant effect on genetic risk, e.g., HLA-DQ in coeliac disease, it might be possible to estimate prevalence using just this variant. However previous work has shown a GRS including the predominant variant as well

as smaller effect variants has better discriminative ability than the predominant variant alone[9].

**Different methods have different advantages.** In most settings, the best approaches are the Means, EMD and KDE methods. The overall performance of these three methods is comparable across different parameters (mixture size, mixture proportional makeup and GRS AUC). At extreme proportions, the KDE method exhibits the smallest bias. A key advantage of the Means method is that it is very straightforward to apply, allowing rapid evaluation of disease prevalence within a cohort. Alternatively, the EMD and KDE methods have the benefit of being able to estimate the prevalence in cases where the Means method cannot be used, e.g., if the reference cohorts have very similar means (Supplementary Fig. 7). Finally, the KDE and EMD methods can be used to test the assumption that the mixture is only composed of two cohorts (Supplementary Note 2).

As noted in the original article by Thomas et al.[7] the Excess method inherently underestimates the proportion of cases because typically both reference cohorts have values below the median value of $R_N$. Taking distinct approaches, the new methods eliminate this inaccuracy and even with decreasing genetic discrimination, these are still interpretable, reflecting the improved generalizability of these methods. We note that the Excess method could be modified to improve its accuracy (e.g., by choosing another quantile rather than the median) but these changes would require case-by-case fine-tuning and at best achieve equivalence to the proposed alternative methods.

**Utility of using polygenic approaches to estimate prevalence within a group**
*Prevalence.* We highlight the clinical utility of the presented concept with a clinical question around the prevalence of potentially undiagnosed coeliac disease within a cohort adhering to a gluten-free diet. This question would be unanswerable using the traditional clinical approach of endoscopy to confirm the coeliac disease, as once observing a gluten-free diet findings are often normal[16]. We showed the prevalence of coeliac disease determined genetically and reported clinically were comparable, suggesting that there is no undiagnosed coeliac disease within this gluten-free cohort. Whilst this finding is not unexpected, it could not be robustly shown before and highlights the general applicability of the proposed framework to quantitatively answer novel and difficult-to-answer questions.

*Defining clinical characteristics of a genetically defined subgroup.* A further advantage of the proposed methodologies over traditional clinical classification arises from the fact that clinical characteristics are not used to define cases. It is therefore possible to estimate both binary and continuous clinical characteristics of the genetically defined disease group within the mixture cohort. Using BMI as an example:

$$\bar{x}_C^{BMI} = \frac{\bar{x}_M^{BMI} - \hat{p}_N \bar{x}_N^{BMI}}{\hat{p}_C} \qquad (1)$$

where $\hat{p}_N$ and $\hat{p}_C$ represent the estimated proportions and $\bar{x}_N^{BMI}$, $\bar{x}_M^{BMI}$ and $\bar{x}_C^{BMI}$ represent the mean BMI of each of the non-cases, mixture and cases (disease) groups respectively. This approach was used in[7] to show rates of Diabetic Ketoacidosis to be the same in T1D diagnosed above and below 30 years of age. We note that all the same limitations of the Means method apply. The EMD and KDE methods could allow for reconstruction of the full distribution of the clinical characteristic, however, evaluation of this approach is beyond the scope of this study.

*Testing of proposed clinical discriminators.* Another utility of these genetic discrimination techniques is to test the performance of clinical classification criteria and allow more precise stratification of a population. Whilst the increasing availability of population datasets generated from routinely collected data allow large-scale population analysis, robust classification can become more difficult leading to bias which is difficult to quantify[6]. Treating the clinically defined cohort as a mixture would allow rapid estimation of the correctly and incorrectly classified proportions within the cohort, thus allowing for bias adjustment and optimisation of classification criteria.

**Cautions.** The use of genetic data in the context of genetic stratification means certain assumptions must hold true for the estimates to be valid. The same assumptions required for Mendelian randomisation[1,17] should be met here. Key to the accuracy of estimates is the equivalence assumption which states that cases and non-cases in the mixture reflect their respective reference cohorts. The importance of meeting this assumption and the implications if it is not met, are highlighted by our example of a raised T2DGRS for an enriched reference T2D population from the WTCCC[8]. For this reason, to help ensure equivalence is maintained we recommend these methods are used in subsets of a cohort allowing reference cases and non-cases to be derived from the same dataset. If these methods are to be used with reference cohorts from different datasets, as was done previously[7], the equivalence assumption should be rigorously tested prior to analysis. This must initially involve a detailed assessment of the selection criteria for the mixture and reference cohorts and available literature, followed by comparison of the GRS between definite non-cases and cases from within the mixture and their respective references.

Careful GRS comparison between the reference cohorts and definite cases and non-cases from the mixture will also help mitigate any potential impact of unrecognized genotype–phenotype interactions, which may arise when selecting subgroups. This is highlighted by Supplementary Fig. 6 showing a reduction in performance of estimates with higher disease prevalence owing to a subtle difference in GRS between type 2 diabetes cases with and without microalbuminuria. We recommend careful investigation for overlapping genetic associations and pleiotrophy using standard Mendelian Randomisation approaches[17,18]. Genotype–phenotype interaction is also relevant to think about the criteria used to originally select cases and controls in the genome-wide association study (GWAS) from which a GRS is derived from, as cases may have been enriched to improve variant discovery. However, this will

have minimal impact on the methods' estimates provided that genetic equivalence has been maintained between reference and mixture cases and controls. Clearly, this would not be the case if the enriched GWAS population was used as the case reference population, as highlighted by our type 2 diabetes example above.

Finally, all our methodologies assume that the mixture consists of only the two genetic reference cohorts such that $p_C + p_N = 1$. Both, the EMD and KDE methods provide a way to check if this mixture assumption is satisfied. In the case of the EMD method we could use the independent estimates of $\hat{p}_N$ and $\hat{p}_C$ to check how much their sum deviates from 1. For the KDE method, the validation could be based on the residuals of the least-square fitting procedure. To check if the deviation from $p_C + p_N = 1$ is significant we again suggest the use of the bootstrap methodology. We present some details and an example of such checks in the supplementary information, however a detailed analysis of this aspect of the proposed methodology is beyond the scope of this paper.

In summary, we propose novel approaches that use population distributions of GRS to estimate disease prevalence. We show that the proposed Means, EMD and KDE approaches improve upon the existing Excess method, performing similarly across different mixture cohorts, with robust estimates possible even when using GRS with reduced discriminative ability. Utilising these concepts will allow researchers to gain novel unbiased insights into polygenic disease prevalence and clinical characteristics, unhampered by clinical classification criteria.

## Methods

**Participants.** **Type 1 diabetes cases:** Cases ($n = 1,963$) were taken from the WTCCC[8]. The WTCCC T1D patients all received a clinical diagnosis of T1D at <17 years of age and were treated with insulin from the time of diagnosis.

**Type 2 diabetes cases:** Cases ($n = 1,924$) were taken from the WTCCC[8]. The WTCCC T2D patients all received a clinical diagnosis of T2D.

**Clinical examples.**

(1) **Coeliac Disease**
   **Coeliac disease reference cases:** Cases ($n = 12,018$) Cases consisted of those from a combination of European studies. Cases were diagnosed according to standard clinical criteria, including compatible serology and small intestinal biopsy[19].
   **Coeliac non-cases:** Non-cases ($n = 12,000$) a cohort was randomly selected from those within the UK biobank (total $n = 366,326$) defined as unrelated individuals of white European descent without a diagnosis of coeliac disease and not reporting a gluten-free diet.
   **Gluten-free diet:** Gluten-free cases ($n = 12,757$) were taken from unrelated individuals of white European descent in the UK biobank reporting adherence to a gluten-free diet.
   **Reported coeliac cases in biobank:** Coeliac disease cases ($n = 1,772$) were defined based on self-reported questionnaire answers and/or an ICD10 record from hospital episode statistics data.

(2) **Microalbuminuria**
   **Type 2 diabetes reference cases:** Cases ($n = 13,268$) were defined as non-insulin-treated participants of White European descent either self-reporting diabetes or with an HbA1C ≥ 48 mmol mol$^{-1}$ at recruitment to UK Biobank without microalbuminuria.
   **Type 2 diabetes non cases:** Non cases ($n = 10,000$) were randomly selected from all participants ($n = 339,385$) of white European descent without microalbuminuria not self-reporting diabetes and with an HbA1C < 48 mmol mol$^{-1}$ at recruitment to UK Biobank.
   **Microalbuminuria cases:** Cases ($n = 17,868$) were taken from unrelated individuals of white European descent in the UK Biobank. We used the albumin creatinine ratio (ACR) calculated from the baseline assessment. In UKBB, a continuous measure of ACR was derived using urinary measures of albumin and creatinine. Microalbuminuria was defined based on international cut-offs ≥ 2.5 mg mmol$^{-1}$ in males and ≥ 3.5 mg mmol$^{-1}$ in females. Any self-reported insulin-treated diabetes cases were excluded as evaluating the proportion of type 2 diabetes cases.
   **Micro-albuminuria cases with type 2 diabetes:** Cases ($n = 2,509$) were defined as white European participants with type 2 diabetes and microalbuminuria as defined by the aforementioned criteria.

(3) **Glaucoma**
   **Type 2 diabetes reference cases:** Cases ($n = 15,128$) were defined as non-insulin-treated participants of White European descent either self-reporting

diabetes or with an HbA1C ≥ 48 mmol mol$^{-1}$ at recruitment to UK Biobank without either self-reported glaucoma or a glaucoma code in hospital episode statistic data.

**Type 2 diabetes non cases:** Non cases ($n = 10{,}000$) were randomly selected from all participants ($n = 345{,}534$) of white European descent without glaucoma not self-reporting diabetes and with an HbA1C < 48 mmol mol$^{-1}$ at recruitment to UK Biobank.

**Glaucoma cases:** Cases ($n = 9{,}857$) were taken from unrelated individuals of white European descent in the UK Biobank self-reporting glaucoma or with a glaucoma code in the hospital episode statistic data. Any self-reported insulin-treated diabetes cases were excluded.

**Glaucoma cases with type 2 diabetes:** Cases ($n = 650$) were defined as white European participants with type 2 diabetes as defined by the aforementioned criteria and self-reporting glaucoma or a glaucoma code in hospital episode statistic data.

**Calculating GRS.** T1DGRS: The T1DGRS was generated using published variants known to be associated with the risk of T1D. We generated a 30 SNP T1D-GRS from variants present in the WTCCC cohort. We followed the method as described by Oram et al.[2] using tag variants rs2187668 and rs7454108 to determine HLA DR haplotype and ascertain the HLA-haplotype component of each individual's score[20]. This was added to the score of the remaining variants, generated by summing the effective allele dosage of each variant multiplied by the natural log (ln) of the odds ratio.

T2DGRS: The T2DGRS was generated using published variants known to be associated with risk of T2D[21]. We generated a 77 SNP T2D-GRS in both the WTCCC cohort and UK Biobank consisting of variants present in both data sets and with high imputation quality (R2 > 0.4). The AUC (0.65) for discriminating T1D and T2D was calculated within the study as this 77 SNP GRS was created specifically to allow comparison between WTCCC cohort and UK Biobank. The score was generated by summing the effective allele dosage of each variant multiplied by the natural log (ln) of the odds ratio.

CDGRS: The 46 SNP coeliac GRS was generated using published variants known to be associated with risk of Coeliac disease[19,22,23]. The log-additive CDGRS was generated using a weight as the natural log of corresponding odds ratios. For each included genotype at the DQ locus, the odds ratio was derived from a case-control dataset[19]. For each non-HLA locus, odds ratios from existing literature were used, and each weight was multiplied by individual risk allele dosage[9,19].

**Excess method.** Following on from the previous work[7], the Excess method calculates the reference proportions in a mixture cohort according to the difference in expected numbers either side of the reference cohort's median. The reference median in question was taken to be the closest to the mixture cohort's median. The proportion was then calculated according to: $\hat{p}_C = \left| \frac{\#\{x > m\} - \#\{x \leq m\}}{n} \right|$, where $m$ is the median of the reference cohort, $n$ is the size of the mixture cohort and $x$ is an individual participant in the mixture cohort, hence $\#\{x > m\}$ represents the number of cases above the median and $\#\{x \leq m\}$ represents the number of cases below the median.

**Means method.** The mean GRS were computed for each of the two reference cohorts and the mixture population. The proportions of the two reference cohorts were then calculated according to the normalised difference of the mixture cohorts's mean ($\mu_M$) and the means of the two reference cohorts ($\mu_{R_C}$ and $\mu_{R_N}$): $\hat{p}_C = \left| \frac{\mu_M - \mu_{R_N}}{\mu_{R_C} - \mu_{R_N}} \right|$. If the mean of the mixture cohort is bigger (or smaller) than both means of the reference cohorts then the estimate is defined as 1 (or 0) depending on the closest reference mean.

**Earth Mover's Distance (EMD) method.** Intuitively, the Earth Mover's Distance (EMD) is the minimal cost of work required to transform one 'pile of earth' into another; with each 'pile of the earth' representing a probability distribution. Mathematically, the (EMD) is a Wasserstein distance and has been widely used in computer and data sciences[24,25]. For univariate probability distributions, the EMD has the following closed-form formula:[26]

$$\mathrm{EMD}(\mathrm{PDF}_C(z), \mathrm{PDF}_N(z)) = \int_z |\mathrm{CDF}_C(z) - \mathrm{CDF}_N(z) \mathrm{d}z| \qquad (2)$$

Here, PDF$_C$ and PDF$_N$ are two probability density functions with support in set Z, and cumulative density functions, CDF$_C$ and CDF$_N$, are their respective cumulative distribution functions.

To compute the EMD, we first find the experimental CDFs of GRS for each of the two reference cohorts and the mixture cohort. These CDFs are then interpolated at the same points for each distribution, with the points being the centres of the bins obtained when applying the Freedman-Diaconis rule[27] to the combined reference cohorts (such that $h = 2 \frac{\mathrm{IQR}}{n^{1/3}}$). As a support set, we take an interval bounded by the minimum and maximum value of the GRS in all three

cohorts. The proportions were then calculated as:

$$p_x^{\mathrm{EMD}} = 1 - \mathrm{EMD}(\mathrm{R}_x, \widetilde{\mathrm{M}}) / \mathrm{EMD}(\mathrm{R}_C, \mathrm{R}_N),$$

where x is either C or N. Since the two estimates are independent, deviation of their sum from one, $\left| p_C^{\mathrm{EMD}} + p_N^{\mathrm{EMD}} - 1 \right|$ can be used to test the assumption that $p_C + p_N = 1$, dispersion of the deviation can be computed during bootstrapping and compared with the value observed in the analysed cohort. However, under the assumption that $p_C + p_N = 1$, we adapted the method by taking the average of the estimated proportions as follows:

$$\hat{p}_C = \frac{p_C^{\mathrm{EMD}} + (1 - p_N^{\mathrm{EMD}})}{2} = \frac{\mathrm{EMD}(\mathrm{R}_C, \mathrm{R}_N) + \mathrm{EMD}(\mathrm{R}_N, \widetilde{\mathrm{M}}) - \mathrm{EMD}(\mathrm{R}_C, \widetilde{\mathrm{M}})}{2 \cdot \mathrm{EMD}(\mathrm{R}_C, \mathrm{R}_N)}.$$

**Kernel Density Estimation (KDE) method.** Individual GRS were convolved with Gaussian kernels, with the bandwidth set to the bin size obtained when applying the Freedman-Diaconis rule[27] in the same way as for the EMD method. This forms two reference distribution templates and a mixture template, KDE$_C$, KDE$_N$ and KDE$_M$ for each dataset. A mixture model was then defined as the weighted sum of the two reference templates (with both weights initialised to 1). This model was then fit to the mixture template (KDE$_M$) with the Levenberg-Marquardt (Least Squares) algorithm[28], allowing the weights ($w_C$ and $w_N$) to vary. The proportions were then calculated according to: $\hat{p}_C = \frac{w_C}{w_C + w_N}$. Admissible values of the weights are limited to the [0, 1] interval.

**Simulated mixtures.** To simulate a range of real-world scenarios, we constructed artificial mixture cohorts by randomly sampling with replacement GRS from the reference cohorts of cases ($\mathrm{R}_C$) and non-cases ($\mathrm{R}_N$) in specified proportions, $p_C$ and total mixture sizes, $n$. To construct the mixtures, we use the WTCCC[8] T1D ($n = 1{,}963$) and T2D ($n = 1{,}924$) data. We used half of the available samples as reference cohorts (first $n = 982$ and $n = 962$ points, respectively) and the other half (last $n = 981$ and $n = 962$, respectively) is a hold-out set used to construct the mixtures. To obtain any required mixture size we sampled with replacement from the hold-out data.

For the heatmaps, Fig. 3 and Supplementary Figs. 2–4, the proportion and cohort size were systematically varied, with $p_C$ ranging from 0 to 1 in 0.01 (1%) steps while $n$ ranged from 100 to 2,500 in steps of 100 samples. All four methods were applied to each combination of these parameters. At each point in the parameter space, we estimated the prevalence ($\hat{p}_C$) and its confidence interval and then compared it with the model proportion ($p_C$) used to generate them.

Figure 3 (top row) illustrates how the randomness of the simulated mixture cohort affects the variability of each method's estimates. This variability reflects the randomness that is inherently present in the mixture cohort. Supplementary Fig. 2 shows how this variability decreases for more discriminative GRS, while Supplementary Note 1 and Supplementary Figs. 3–4 compare the performance of the methods once the randomness of the composition of the mixture cohort is eliminated.

For Supplementary Figs. 5–6, we used the GRS of the T2D cases and non-cases in the mixture cohort, $\widetilde{\mathrm{M}}$, to construct (random sampling with replacement) 21 artificial mixture distributions, ($n = 2{,}500$, each) with prevalence of T2D varying from 0 to 100% (with of 5% step). To estimate the proportions in the constructed mixture cohorts we used reference cohorts as specified in Clinical examples section above.

**Synthetic GRS data.** To generate synthetic GRS in Fig. 4 and Supplementary Fig. 7 we used pseudorandom number generators. As references, we used two samples ($n = 2{,}000$, each) from normal distributions with mean 0 and standard deviation 1, N(0, 1); the means and standard deviations of the reference samples were $\mu = -0.002$, $\sigma = 0.999$ and $\mu = -0.008$, $\sigma = 1.001$. The reference samples are generated only once. To change the AUC for the reference samples, we added a value to one distribution of them to change its mean. The mixtures are generated using different pseudorandom number generators for each proportion ($p_C$) and AUC value. For example, to generate mixture with $n = 5{,}000$, $p_C = 0.1$ and AUC = 0.7 we: (1) draw 500 samples from N(0, 1) and (2) we draw $n = 4{,}500$ samples from N(0, 1) and add 0.74 to them. The mixture and reference samples are generated separately.

**Varying mixture size.** To investigate the dependence of the width of the CIs on the mixture size (Fig. 4c) and find the minimum mixture size required for CIs width <0.1 (Table 1) we used the same synthetic GRS distributions with $p_C = 0.1$ as described above. For the Excess, Means and the EMD methods, we varied mixtures sizes between 100 and 10,000 (30,000 for AUC < 0.7) with a step of 100 points. Since the KDE method is more computationally expensive, we tested mixture sizes between 100 and 6,500 with a step of 100 points and between 7,000 and 10,000 (40,000 for AUC = 0.6, 30,000 for AUC = 0.65) with a step of 500 points. For each considered mixture size, we repeated the estimation of the CIs 100 times. We disregarded estimates for which the CIs do not include $p_C = 0.1$. As the minimum mixture size, we took a median of the mixture sizes (over the 100 runs) at which we first observed the CI width <0.1.

**Calculating confidence intervals**. In order to estimate confidence intervals and any systematic bias of the methods, we used Monte Carlo[11] and bootstrap methods[12,29]. We combined the two approaches to capture variability of the estimate resulting from the mixture size and features of the reference distributions.

First, we stochastically modelled the process of generating the mixture. To do so, we modelled $N_M$ new mixtures, by sampling with replacement from the reference cohorts. Each modelled mixture has the same size as the original cohort and the composition given by the initial estimate $\hat{p}_C$ based on the original mixture. For example, if the original cohort has 1,000 values and the estimate was $\hat{p}_C = 0.3$ then each modelled mixture would contain 300 values sampled with replacements from the cases reference sample ($R_C$) and 700 values from the non-cases reference sample ($R_N$). Next, we resampled each of the $N_M$ new mixtures generating $N_B$ bootstrap samples, see also Supplementary Fig. 8.

Following, chapters 2 and 5 from ref. [12] we used all the $N_M \cdot N_B$ cohorts to compute the bias and confidence intervals of the estimate. The systematic median bias of the method is defined as a difference between $\mathrm{med}(\{\{p'_C\}_B\}_M)$ the median value of the $N_M \cdot N_B$ bootstrapped estimates of $p'_C$ and the estimate $\hat{p}_C$:

$$B = \mathrm{med}(\{\{p'_C\}_B\}_M) - \hat{p}_C. \tag{3}$$

We used bias corrected and accelerated bootstrap confidence intervals (BCa CI) which we computed as described in ref. [30]. Bootstrap confidence intervals assume that the spread of the distribution of the bootstrap estimates $p'_C$ can be used to estimate the CI. The BCa CI take into account median bias and skewness (acceleration) of the distribution of the bootstrap estimates $p'_C$ and allows calculation of corrected quantiles representing a chosen confidence level, $\alpha$.

Throughout this section $\Phi$ is a normal standard ($\mu = 0, \sigma = 1$) CDF and $\Phi^{-1}$ is its inverse and $\mathscr{T}_n^{-1}$ is an inverse CDF of a Student's t-distribution with $n$ degrees of freedom.

The computation takes the following steps:

1. Estimate the median bias correction factor $z_0$:

$$z_0 = \Phi^{-1}\left(\frac{\#(p'_C \le \hat{p}_C)}{N_M \cdot N_B}\right). \tag{4}$$

2. Estimate the acceleration correction factor $\hat{a}$:

$$\hat{a} = \frac{1}{6} \frac{\sum_{i=1}^n U_i^3}{\left(\sum_{i=1}^n U_i^2\right)^{3/2}}. \tag{5}$$

where $U_i$ values are calculated using the jackknife influence function:

$$U_i = (n-1)(\hat{p}_C - \hat{p}_i), \tag{6}$$

here $\hat{p}_i$ is an estimate based on the reduced mixture sample $\tilde{M}_i = (GRS_1, GRS_2, \dots, GRS_{i-1}, GRS_{i+1}, \dots GRS_n)$ with score $i$ removed.

3. To counteract the narrowness bias we additionally expand the confidence level[29]

$$\alpha' = \Phi\left(\mathscr{T}_{n-1}^{-1}(\alpha)\sqrt{(n/(n-1))}\right). \tag{7}$$

4. Use bias and acceleration factors to compute the BCa confidence levels:

$$\alpha_{BCa}(\alpha) = \Phi\left(z_0 + \frac{z_0 + \Phi^{-1}(\alpha')}{1 - \hat{a} \cdot (z_0 + \Phi^{-1}(\alpha'))}\right). \tag{8}$$

5. Take $\alpha_{BCa}(\alpha/2)$ quantile of the $p'_C$ samples to obtain the lower confidence limit $CI_L$ and $\alpha_{BCa}(1 - \alpha/2)$ quantile to obtain the upper confidence limit $CI_U$. If the median bias is very strong the BCa CI are undefined. For example, if the $\hat{p}_C$ is outside of the range of the distribution of the bootstrap estimates $p'_C$, $z_0$ is infinite and both limits of the CIs are equal to the maximum or minimum value the $p'_C$ samples.

**Reporting summary**. Further information on research design is available in the Nature Research Reporting Summary linked to this article.

## Data availability

UK Biobank data can be obtained after completing an online application, see details at http://www.ukbiobank.ac.uk/using-the-resource/ Wellcome Trust Case Control Consortium genotype data can be obtained through by application to the Wellcome Trust Case Control Consortium Data Access Committee. The procedure is described in more detail at https://www.wtccc.org.uk/info/access_to_data_samples.html.

## Code availability

The Distribution Proportion Estimation software (v1.0.0) used to analyse the data was developed and tested in Python 3.8.2 and Matlab release 2020b (that includes other algorithms mentioned in the manuscript). The Distribution Proportion Estimation software (v1.0.0) implementing these methods is archived at https://doi.org/10.5281/zenodo.5512651. The code is open-source and available under version-control here: https://github.com/bdevans/DPE.

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

## Acknowledgements

This research has in part been conducted using the UK Biobank Resource. The authors would like to acknowledge the use of the University of Exeter High-Performance Computing (HPC) facility in carrying out this work. We are grateful to Jack Bowden for his comments on the manuscript.

## Author contributions

Manuscript writing: B.D.E., N.J.T., P.S., K.T.A. Method development: B.D.E., P.S., N.J.T., A.T.H., R.J.O., K.T.A. Data acquisition and coding: N.J.T., S.S., R.K., S.J., M.N.W. Simulation implementation, running and analysis: B.D.E., P.S. Discussion of results and manuscript editing: All authors. Project coordination: K.T.A., N.J.T.

## Funding

B.D.E. and P.S. acknowledge that this work was generously supported by the Wellcome Trust Institutional Strategic Support Awards (WT204909MA and 204909/Z/16/Z respectively). K.T.A. gratefully acknowledges the financial support of the EPSRC via grants EP/N014391/1 and EP/T017856/1. N.J.T. is funded by an NIHR Academic Clinical Fellowship and undertook the research as part of a Wellcome Trust funded secondment within the translational research exchange at Exeter University (WT204909MA and 204909/Z/16/Z respectively). S.A.S. is supported by a Diabetes UK PhD studentship (17/0005757). M.N.W. is supported by the Wellcome Trust Institutional Support Fund (WT097835MF). R.A.O. is funded by a Diabetes UK Harry Keen Fellowship (16/0005529). S.E.J. is funded by an MRC grant. A.T.H. is supported by the NIHR Exeter Clinical Research Facility and a Wellcome Senior Investigator award and an NIHR Senior Investigator award. The views expressed are those of the authors and not necessarily those of the NHS, the NIHR or the Department of Health.

## Competing interests

The authors declare no competing interests
