## [Peer Review File · Nature Communications]

Estimating disease prevalence in large datasets using genetic risk scoresReviewers' Comments:

Reviewer #1:

Remarks to the Author:

The authors propose in this manuscript that from a disease GRS distribution in a population, and leveraging cases and controls reference GRS distributions, is possible to extract an estimate of the disease prevalence in a population, and benchmark 4 (of which 3 novel) methods to perform such evaluation. The general idea is very attractive, might have a great impact and I was not able to find previous applications of it, except for their previous work extensively referenced and improved in this manuscript. On the other hand the task faces several assumptions and issues that might reduce its impact and applicability, which need to be carefully evaluated. The authors comment and analyze some of these difficulties, but not all and some just in the discussion. See specific comments below.

1. My main concern is that for all the methods to work, it is critical that the cases and non-cases GRS distributions in the target population (mixture) follow respectively the reference GRS distribution for cases and the reference GRS distribution for non-cases. As those reference groups are not included in the mixture population, otherwise the method would be circular, this is not given. I cannot find a mention of this until the discussion but I think the authors should introduce this since the beginning, adding a referenced comment on how much across medically defined groups of individuals such assumption is expected to hold and/or offer a way to assess it.
2. Along the same lines, I think that such method is not applicable using references from a "population" to infer prevalence in another, where "population" means ethnicity or another more fine-grained demographic group. Across them GRS distribution are usually directionally biased, without reflecting a disease prevalence difference, as reported in previous literature (PMID: 28366442, 31155286). Such assumption is mentioned in conclusion lines 363-371, but should be anticipated and properly referenced in the introduction as is a known limitation to the method applicability and not a weakness revealed in this work. Furthermore, to avoid confusion in the reader, I would consider avoiding the word population with the meaning of "set of samples" in title and throughout the text, preferring something like cohort, sample set, or what the authors deem appropriate, because it might mislead the reader to a method applicable across populations.
3. Under the same light I think it would be fundamental to avoid testing the methods on a mixture population composed by the same individuals present in the reference, which appears to be the case in results lines 153-182 and possibly lines 186-214. In the real application it is crucial that reference and mixture are non-overlapping and also non-overlapping with the individuals used to generate the effect sizes that compose the GRS to avoid circularity and over-fitting. This might increase the impact of different sample sizes.
4. The sample set description in general is missing throughout the results section and is just quickly mentioned in the figure captions. I think it should be present in the main text, reported with reference in the results for the reader to follow and described in detail in the methods. For example I could not find which is the size of the reference populations or how the authors simulate a mixture with sample size 5000 when both cases and controls are less than 2000.
5. Another major concern is that, as the authors discuss extensively, there are areas where the methods show limitations, i.e. with prevalences far from 50% and with low AUC, but to my knowledge these are also the areas where most of the real-world cases belong. Yet the main figures do not show such cases, and to have an idea the reader should look at supplementary figure 2, which I like a lot because indeed it contains many details. Therefore I think it would be better to include an exploration of T2D GRS or a similarly powered GRS in the main figures, and moreover to compare with a negative control, like a GRS generated with random effect sizes or a GRS for a trait independent from diabetes. The aim of this operation would be to give a sense to the reader of how much a GRS needs to be predictive to be used with these methods, which I think is important to include in the main text. In the

same scope, I think it would be beneficial to give an idea of where in the range of prevalences the methods start to be reliable. I don't think the results in figure 4 fully answer such questions as they only explore 50% prevalence, they don't make use of real, available, GRS with low predictivity and they might suffer from the problem described in comment 4.

Below you can find other minor comments.

6. The authors mention that the sum of EMD estimates might be smaller than 1 and explain this in the supplementary, they state in conclusion line 354-356 that this happens especially with extreme prevalences but I could not find any result related to that. This could be improved adding more references to the supplementary section, summarizing its findings in the main text, and adding a result that justify conclusion lines 354-356.

7. I was a bit confused about the usage of \sim^* in formulas throughout the text, I would advise a standard usage with no diacritic for the true quantity and \wedge for the relative estimate.

8. The authors should add references for a) AUC in line 199 b) AUC in line 203 c) celiac disease prevalence in line 225 d) celiac disease odds ratios in line 425

9. conclusion lines 343-350: I would appreciate the authors to comment on how this application is affected by the discriminators used to select cases and controls in the GWAS analysis that produced the GRS summary statistics.

10. I would avoid the step definitions in figure 3 and 5 and add them as caption or in the methods.

11. I think figure 5 will be more informative zooming around celiac disease prevalences (e.g. p_c axis from 0 to 0.3)

Reviewer #2:

Remarks to the Author:

The manuscript compares several methods for estimating disease prevalence from distributions of genetic risk scores.

While the idea of estimating disease prevalence based on genetic risk score distributions is in principle interesting, I am not convinced that the approaches presented here are useful in practice, for the following reasons:

1. Some of the proposed methods rely heavily on the shapes of the risk score distributions. However, any genetic risk score that can be called polygenic (i.e. it is not dominated by a very small number of loci) will follow a distribution that is close to normal. An exception to that would be a sample comprised of different subpopulations and a polygenic score which is influenced by uncorrected stratification in the underlying GWAS. In this case, any non-normality of the risk score distribution would more likely be a reflection these biases and of the presence of different subpopulations than of a non-normal distribution of true genetic risk.

2. Most diseases have a prevalence far from 0.5, and most genetic risk scores have an AUC much closer to 0.5 than to 1. Figure 4 and Supplementary Figures 1 and 2 suggest that under these scenarios it is not possible to estimate prevalence with any degree of accuracy.

The coeliac disease example is an interesting use case of the proposed method. In an ascertained subsample, the prevalence of a rare disease might be high enough to get useful estimates. However,

the genetic risk for coeliac disease is dominated by the DQ locus in the HLA region, which is probably the reason for the very unusual bimodal GRS distribution of the mixture population. This raises the question if there is any advantage in computing polygenic scores over simply tabulating the DQ genotypes in each of the groups and comparing the relative counts in order to estimate prevalence.

While the assumption of a non-normal polygenic score distribution is the largest problem in my view, there are other issues which might affect all proposed methods, and which have not been investigated in the manuscript. For example, how do population stratification and genetic heterogeneity between the GWAS cohort and the prediction cohorts affect the prevalence estimates? How does case ascertainment affect prevalence estimates? (Does it matter if the true prevalence is 1%, but only the most severe cases were recruited for the GWAS?) What is the effect of different genetic architectures? Investigating these issues may not be necessary if there was a clear demonstration that these methods allow accurate prevalence estimates regardless of potential issues like that. But in my opinion, the coeliac disease example falls short of this because the genetic score distribution seems to track one specific locus and is not representative of typical polygenic scores.

Reviewer #3:

Remarks to the Author:

In this study, Evans et al compare different methods to estimate disease prevalence from the distribution of polygenic scores in a sample of homogeneous ancestry. I found the study not really clear in the presentation of its results and had to wait until the discussion to see somewhat clear conclusions. I think there is limited added value over the study from Thomas et al. (ref. 5) and therefore I don't see the novelty and why that study would interest a wide audience. I do not recommend it for publication in Nature Communications.

Response to reviewers' comments:

Response in red

Reviewer #1 (Remarks to the Author):

The authors propose in this manuscript that from a disease GRS distribution in a population, and leveraging cases and controls reference GRS distributions, is possible to extract an estimate of the disease prevalence in a population, and benchmark 4 (of which 3 novel) methods to perform such evaluation. The general idea is very attractive, might have a great impact and I was not able to find previous applications of it, except for their previous work extensively referenced and improved in this manuscript. On the other hand the task faces several assumptions and issues that might reduce its impact and applicability, which need to be carefully evaluated. The authors comment and analyze some of these difficulties, but not all and some just in the discussion. See specific comments below.

Thank you for your positive review and very helpful comments. We are glad you found the general idea attractive, and we agree that it could have a great impact. We feel this potential is highlighted by our previous application evaluating population prevalence of type 1 diabetes, providing novel insights into a clinical question that was previously difficult to answer. We wanted this manuscript to build on this work showing that this concept has excellent general utility as a novel research tool and not just in diseases with extremely strong genetic predisposition like type 1 diabetes. On reflection of the extremely helpful comments from the review process we feel that this message could have been clearer. We have now significantly revised the manuscript and analyses particularly using more clinically relevant parameters in our analyses and figures including disease prevalence of 10% and GRS from AUC 0.6. We feel adding this analyses alongside earlier discussion of the methodological assumptions and ways to test them means the new manuscript highlights the general utility of genetic stratification.

1. My main concern is that for all the methods to work, it is critical that the cases and non-cases GRS distributions in the target population (mixture) follow respectively the reference GRS distribution for cases and the reference GRS distribution for non-cases. As those reference groups are not included in the mixture population, otherwise the method would be circular, this is not given. I cannot find a mention of this until the discussion but I think the authors should introduce this since the beginning, adding a referenced comment on how much across medically defined groups of individuals such assumption is expected to hold and/or offer a way to assess it.

Response:

Thank you for highlighting that this was not sufficiently emphasised throughout. As you say genetic equivalence between cases and non-cases and their respective reference cohorts is a critical assumption for the methods to work. In the re-worked manuscript we have tried to

far more strongly emphasise the key importance of this assumption. We have added a paragraph (page 6 paragraph 2) far earlier in the manuscript (in the genetic stratification summary) outlining both why this assumption is so important and ways of assessing it to ensure it holds true. The key thing for ensuring equivalence is that the criteria used to select the reference and mixture cohorts are not significantly different and this must be assessed before any analysis is undertaken. In addition to outlining the importance of these steps, for further emphasis we have added a real world example for type 2 diabetes, when the assumption did not hold because a specific selection criteria had been used to select type 2 diabetes cases in the reference cohort which was not reflected by cases in the mixture. This meant the GRS distribution of cases in the reference and mixture were significantly different. Comparing the GRS of the known cases in the mixture with reference cases clearly showed that genetic equivalence was not preserved and analysis would have been inaccurate. When the selection criteria were aligned between the reference and study cohorts genetic parity was restored highlighting how cohort selection criteria is the critical factor in ensuring the genetic equivalence assumption is met.

Added to manuscript: Page 6 Paragraph 2, starting line 124

Finally our methods are all based on the assumption that between the reference and mixture cohorts, cases and non-cases are genetically equivalent. This assumption must hold true for estimates to be valid. This is of particular importance when studying different geographical populations where allele frequencies are known to vary [1, 2]. To ensure this equivalence assumption is fulfilled the selection criteria and demographics of the reference and mixture cohorts should be compared to ensure equivalent. In this manuscript all analyses are restricted to white Europeans, reflecting the populations that the reference GRS distributions were derived from. Furthermore where possible the GRS of the reference non-cases (controls) and cases should be compared with the GRS of non-cases and cases within the same population the mixture has been taken from. This could be done, for example, by means of a statistical test appropriate for assessment of the observed GRS distributions. An example of the importance of this and how it can be detected is demonstrated by the T2DGRS for a reference T2D population from the WTCCC [3]. The WTCCC cohort was largely selected based on a positive family history of T2D or early disease onset and is therefore enriched for T2D risk variants. As shown in Supplementary Fig. 1 the distribution of T2DGRS of unselected T2D cases from population data in UK Biobank is significantly lower than the T2DGRS in the WTCCC T2D reference. The T2DGRS in UK Biobank population T2D cases only becomes equivalent to the WTCCC when the same case selection criteria are mirrored. If this WTCCC cohort was used as a reference T2D population when evaluating prevalence of T2D in a cohort in UK biobank, it would have influenced the accuracy of estimates since it is not reflective of an average T2D cohort.

Sup figure 1

Supplementary Figure 1: shows the effect of non-preserved equivalence. BBT2D = Biobank Type 2 diabetes defined as diabetes diagnosed over 30 years of age and non-insulin treated. BB WTCCC T2 is defined as first degree relative with type 1 diabetes diagnosed aged over 30 up to 40 years of age to recreate the WTCCC cohort. The WTCCC T2 cohort is also shown.

2. Along the same lines, I think that such method is not applicable using references from a “population” to infer prevalence in another, where “population” means ethnicity or another more fine-grained demographic group. Across them GRS distribution are usually directionally biased, without reflecting a disease prevalence difference, as reported in previous literature (PMID: 28366442, 31155286). Such assumption is mentioned in conclusion lines 363-371, but should be anticipated and properly referenced in the introduction as is a known limitation to the method applicability and not a weakness revealed in this work. Furthermore, to avoid confusion in the reader, I would consider avoiding the word population with the meaning of “set of samples” in title and throughout the text, preferring something like cohort, sample set, or what the authors deem appropriate, because it might mislead the reader to a method applicable across populations.

Response

We are grateful for you pointing out how our use of population was confusing and think the word cohort is clearer, which we now use throughout the manuscript in its place. We agree the importance of GRS distributions altering in different ethnicity or fine-grained demographic groups should be addressed earlier. We have added this in the genetic stratification summary straight after the introduction (as written out above, Page 6 Paragraph 2) and included your highlighted references which we think illustrate these issues

nicely. We felt this didn't quite fit in the introduction narrative but is now far earlier in the revised manuscript than previously. We would happily move this to the introduction if the editors felt this more appropriate. We agree this is not a weakness inherent to our method so have moved this to a newly created cautions paragraph in the discussion (page 20 paragraph 3). This section highlights key considerations (including point 1 above) before the methods can be applied to ensure equivalence is maintained between reference and mixture cohorts and resultant prevalence estimates are robust.

Included on page 20 paragraph 3, starting line 464

Cautions

The use of genetic data in the context of genetic stratification means certain assumptions must hold true for the estimates to be valid. The same assumptions required for Mendelian randomisation [4, 5] should be met here. Key to the accuracy of estimates is the equivalence assumption which states that cases and non-cases in the mixture reflect the respective reference cohorts. This is particularly important when studying different geographical populations where allele frequencies may vary, leading to an alteration in genetic risk scores across the cohorts [1, 2]. The assumption will also fail where the GRS is different between cases or non-cases in the reference and mixture. This is shown by our example of a higher T2DGRS distribution for a WTCCC reference T2D population compared to T2D cases in UK Biobank, Supplementary Fig. 1. This reflected the WTCCC cohort being enriched for T2D risk variants as selected based on a positive family history of T2D or early disease onset [3]. This highlights the importance of testing the equivalence assumption prior to analysis. This should initially involve detailed assessment of the selection criteria for the mixture and reference cohort and available literature. Furthermore we suggest comparing the GRS of the reference non-cases (controls) and cases with, where available, the GRS of definite non-cases and cases within the same population the mixture has been taken from (e.g., by means of a statistical test appropriate for assessment of the observed GRS distributions).

3. Under the same light I think it would be fundamental to avoid testing the methods on a mixture population composed by the same individuals present in the reference, which appears to be the case in results lines 153-182 and possibly lines 186-214. In the real application it is crucial that reference and mixture are non-overlapping and also non-overlapping with the individuals used to generate the effect sizes that compose the GRS to avoid circularity and over-fitting. This might increase the impact of different sample sizes.

Response:

This is a very valid point – thank you for highlighting it. In the revised manuscript where we have created artificial mixtures, we have now split our data in two, so one half is used to create the mixture and the other is used for the reference populations. Therefore cases are in either reference or mixture which we agree far more realistically reflects real world performance.

For example added Page 5 paragraph 2, starting line 99

To compare the methods under different conditions, the T1DGRS data were split in half to form reference cohorts and an independent hold-out set for generating parameterised mixture cohorts

4a. The sample set description in general is missing throughout the results section and is just quickly mentioned in the figure captions. I think it should be present in the main text, reported with reference in the results for the reader to follow and described in detail in the methods.

Response

The detailed description of the clinical sample sets is in the main methods section at the end of the manuscript. We did this in keeping with the journal style but could provide more detail earlier in the manuscript if the editors would like us to. We have now added a brief sample set description in the summary methods straight after the introduction, as we agree that this makes figure interpretation easier.

Added Page 5 paragraph 1, starting line 90

Clinical sample sets were taken from the following cohorts: T1D (n=1,963) and T2D (n=1,924) from the Wellcome Trust Case Control Consortium (WTCCC) [3], Coeliac disease reference cases (n=12,018) from a combination of European studies [15] with non-cases (controls) and mixture (gluten-free diet) cohorts from UK Biobank (n=12,000 and n=12,757, respectively) [6].

4b. For example I could not find which is the size of the reference populations or how the authors simulate a mixture with sample size 5000 when both cases and controls are less than 2000.

Response:

With regard to generating a sample sizes larger than available data we used sampling with replacement and have now explained and referenced this recognised approach in more detail. This approach was unavoidable given the size of mixture cohorts required to fully demonstrate the generalisability of our methods across a variety of different parameters.

Added Page 5 paragraph 2: starting line 101

In these analyses, mixtures were constructed by sampling with replacement, enabling larger mixture sizes to be used than the size of the hold-out sets from which they were derived.

5. Another major concern is that, as the authors discuss extensively, there are areas where the methods show limitations, i.e. with prevalences far from 50% and with low AUC, but to my knowledge these are also the areas where most of the real-world cases belong. Yet the main figures do not show such cases, and to have an idea the reader should look at supplementary figure 2, which I like a lot because indeed it contains many details. Therefore

I think it would be better to include an exploration of T2D GRS or a similarly powered GRS in the main figures, and moreover to compare with a negative control, like a GRS generated with random effect sizes or a GRS for a trait independent from diabetes. The aim of this operation would be to give a sense to the reader of how much a GRS needs to be predictive to be used with these methods, which I think is important to include in the main text. In the same scope, I think it would be beneficial to give an idea of where in the range of prevalences the methods start to be reliable. I don't think the results in figure 4 fully answer such questions as they only explore 50% prevalence, they don't make use of real, available, GRS with low predictivity and they might suffer from the problem described in comment 4.

Response

Thank you for this comment. We have put a lot of focus into the revised manuscript around this, as we agree it is key and we think this now greatly improves the relevance of our findings. Throughout we have altered the figures to use more clinically relevant parameters. We have moved the T2DGRS heat map into the main manuscript as figure 4 as we agree this does highlight the applicability of these methods using a real world GRS with reduced AUC (0.65). For Fig 3 and the previous Fig 4 (now Fig 5) we have reanalysed and now show results with a minimum disease prevalence of 10%. Where AUC allows methods can be used with even lower disease prevalence ($\approx 5\%$) (Sup fig2). However we also highlight that diseases rare at a population level can still be studied by enriching for cases using careful choice of mixture criteria. For example type 1 diabetes population prevalence is 0.5% but within insulin treated diabetes cases has a prevalence of 25% and this variable will capture all type 1 diabetes cases.

With regard to when the methods start to be reliable in terms of genetic predisposition (AUC) we have tried hard to make this clearer. A key point is that the AUC of the GRS, the size of the mixture cohort and the prevalence of the disease all interact to affect estimate accuracy. We now explore these factors and their interaction in far more depth within clear titled sections within the results and discussion particularly to address the key question of how discriminant a GRS needs to be. We have added Table 1 to show the minimum number of cases for each method that allows robust estimates around a prevalence of 10% with a CI $\pm 5\%$ using AUC ranging upwards from 0.6. This clearly shows that away from severe extremes of low prevalence or AUC the reduction in accuracy can be mitigated by increasing cohort size. Whilst 25,500 initially feels like a large cohort a disease with 10% prevalence means just 2,550 cases required with the remaining majority controls and given the size of population data sets available this is readily achievable.

We feel these changes to the manuscript highlight the generalisability of our approach even in situations with reduced genetic predisposition. As you say it is worth acknowledging that there will be disease with minimal (AUC < 0.6) or no (AUC=0.5) polygenic risk where these approaches are probably not appropriate. We have therefore not shown a separate negative control as feel this might be confusing but feel Fig 5 does show AUC of 0.5 to 0.6 and how unusable estimates are in this situation.

Added titles

Results: page 12 paragraph 1 line 250: **How predictive does a genetic risk scores need to be?**

Discussion: page 18 paragraph 2 page 383: **Estimates remain robust in diseases with less discriminative genetic risk scores**

Table 1 page 13 page 273

Method/A	UC	0.6	0.65	0.7	0.75	0.8	0.85	0.9
Excess	-	-	-	-	-	3200	3100	3100
	Q25: -	Q25: -	Q25: -	Q25: -	Q25: -	Q25:300	Q25:290	Q25:300
	Q75: -	Q75: -	Q75: -	Q75: -	Q75: -	0	0	0
	87/100	62/100	42/100	19/100	Q75:340	Q75:330	Q75:320	Q75:320
					0	0	0	0
					13/100	2/100	3/100	
Means	26500	11500	6200	3700	2500	1700	1100	
	Q25:258	Q25:111	Q25:600	Q25:360	Q25:240	Q25:160	Q25:110	
	50	00	0	0	0	0	0	
	Q75:273	Q75:117	Q75:637	Q75:390	Q75:252	Q75:170	Q75:120	
	00	00	5	0	5	0	0	
4/100	1/100	1/100	0/100	3/100	4/100	1/100		
EMD	25500	10800	5700	3400	2200	1500	1000	
	Q25:246	Q25:104	Q25:555	Q25:330	Q25:210	Q25:140	Q25:100	
	00	00	0	0	0	0	0	
	Q75:263	Q75:112	Q75:600	Q75:360	Q75:230	Q75:150	Q75:100	
	00	00	0	0	0	0	0	
0/100	0/100	0/100	1/100	0/100	2/100	0/100		
KDE	38250	17000	9000	5500	3400	2100	1300	
	Q25:370	Q25:160	Q25:850	Q25:530	Q25:330	Q25:210	Q25:130	
	00	00	0	0	0	0	0	
	Q75:395	Q75:180	Q75:912	Q75:570	Q75:350	Q75:220	Q75:140	
	00	00	5	0	0	0	0	
54/100	17/100	15/100	4/100	3/100	0/100	0/100		

Below you can find other minor comments.

6. The authors mention that the sum of EMD estimates might be smaller than 1 and explain this in the supplementary, they state in conclusion line 354-356 that this happens especially with extreme prevalences but I could not find any result related to that. This could be improved adding more references to the supplementary section, summarizing its

findings in the main text, and adding a result that justify conclusion lines 354-356.

Response:

When re-writing we decided to remove this paragraph as we felt it confused rather than enhanced our general message and therefore was beyond the scope of this paper.

7. I was a bit confused about the usage of \sim^* in formulas throughout the text, I would advise a standard usage with no diacritic for the true quantity and \wedge for the relative estimate.

Response:

We agree and have, changed it as suggested - no diacritic for the true quantity and \wedge for the relative estimate

8. The authors should add references for a) AUC in line 199 b) AUC in line 203 c) celiac disease prevalence in line 225 d) celiac disease odds ratios in line 425.

Response.

Thank you we have referenced by a) added (line 185) b) This 77SNP GrS was generated specifically for this analysis so not previously generated but we have made this clear in text see below c) this coeliac prevalence in biobank was calculated as part of our analysis, we have clarified the text and added the UK Biobank reference for completeness line 322). D) added line 546

8b added:

Page 12 paragraph 1 line 263

AUC 0.65 (calculated in this study))

Page 22 final paragraph line 537

The AUC (0.65) for discriminating T1D and T2D was calculated within the study as this 77 SNP GRS was created specifically to allow comparison between WTCCC cohort and UK Biobank.

9. conclusion lines 343-350: I would appreciate the authors to comment on how this application is affected by the discriminators used to select cases and controls in the GWAS analysis that produced the GRS summary statistics.

Response:

Thank you for this interesting comment which highlights that with regard to the GWAS from which the GRS has been derived we are assuming that the GWAS has been conducted using “pure” cases and controls. It is possible that misclassified cases and controls might have inadvertently been included in the original GWAS the GRS was derived from. This is likely to be minimal and at worst negligibly reduce the observed odds ratio of SNPS used thereby reducing the observed AUC of the GRS generated. This should not affect estimates made in the context of this application other than a subtle reduction in precision of estimates in line with reduced AUC as previously discussed.

Another possible related caution is that case selection for GWAS could have been applied to aid variant discovery in situations where there is strong genotype phenotype interaction e.g. using young disease age of onset. The caution around this largely links back to your first point on genetic equivalence between reference and mixture cases and controls and is therefore only relevant if that same discovery GWAS cohort is then used as the reference cohort for our methods. In this situation genetic equivalence would not hold true as in our Type 2 diabetes example. If the GRS (derived from an enriched GWAS) is used for separate reference populations where the selection criteria is the same as the mixture then this will apply equally to all cohorts and not impact estimates. We have added a short summary of this to cautions as we feel this key point is relevant to all applications proposed and not just the one highlighted.

Added: Page 20 paragraph 3 starting line 481

Genotype phenotype interaction is also relevant to thinking about the criteria used to originally select cases and controls in the genome wide association study (GWAS) from which a GRS is derived as cases may have been enriched to improve variant discovery. However this will have minimal impact on method estimates as long as genetic equivalence has been maintained between reference and mixture cases and controls. Clearly this would not be the case if the enriched GWAS population was used as the case reference population as highlighted by our Type 2 diabetes example above.

10. I would avoid the step definitions in figure 3 and 5 and add them as caption or in the methods.

response:

Response:

We agree and have removed.

11. I think figure 5 will be more informative zooming around celiac disease prevalences (e.g. p_c axis from 0 to 0.3)

Response:

We agree and have done this.

Reviewer #2 (Remarks to the Author):

The manuscript compares several methods for estimating disease prevalence from distributions of genetic risk scores.

While the idea of estimating disease prevalence based on genetic risk score distributions is in principle interesting, I am not convinced that the approaches presented here are useful in practice, for the following reasons:

1. Some of the proposed methods rely heavily on the shapes of the risk score distributions. However, any genetic risk score that can be called polygenic (i.e. it is not dominated by a very small number of loci) will follow a distribution that is close to normal. An exception to that would be a sample comprised of different subpopulations and a polygenic score which is influenced by uncorrected stratification in the underlying GWAS. In this case, any non-normality of the risk score distribution would more likely be a reflection these biases and of the presence of different subpopulations than of a non-normal distribution of true genetic risk.

Response

Thank you for this comment. On reflection we can see that the original manuscript gave the impression that a non-normal GRS distribution was required. In fact, all three new proposed methods work extremely well with normally distributed GRS, the situation, as you point out found with the majority of diseases with polygenic inheritance. Our results show that the means method, which uses a single summary statistic and is therefore agnostic to the GRS distribution, performs equally well as the EMD and KDE method in the majority of scenarios and all situations using normally distributed GRS. In the re-worked manuscript we have tried to make this clear with the key results derived from using normally distributed GRS including figure 4 (using T2DGRS), figure 5 and table 1. As you say, diseases with genetic predisposition dominated by a very small number of high-risk loci will have non normal GRS distributions, for example in disease with strong HLA associations like type 1 diabetes. Whilst this additional information favours the EMD and KDE methods, in this scenario the performance improvement is minimal and the means method still provides robust estimates.

Added

Results page 12 paragraph 1 starting line 264

It is worth noting that these examples use normal distributions with equal standard deviations, representing the majority of polygenic GRS. The performance of the EMD and

KDE methods, which are non-parametric, will be enhanced relative to the Means method when the reference cohort means are close but there are differences in other characteristics of the GRS distributions (e.g., skewness, kurtosis or multi-modality). This is the case, for example, when certain genetic variants predominate leading to a skewed GRS distribution e.g., in autoimmune diseases such as T1D with strong HLA association.

Discussion page 18 paragraph 2 starting line 390

Performance of the Means, EMD and KDE methods is very good in the case of normal GRS distributions with equal standard deviations e.g., diseases with polygenetic risk arising from a large number of causal variants, each with tiny effects e.g., T2D. In diseases where certain variants predominate e.g., HLA in autoimmune disease, the GRS will be skewed to account for this e.g., T1D. In this instance the EMD and KDE methods will be more accurate, as they are able to utilise the unequal skewness (or other properties such as standard deviations or kurtosis) even when the means of the reference distributions are close, see Supplementary Fig. 5.

2. Most diseases have a prevalence far from 0.5, and most genetic risk scores have an AUC much closer to 0.5 than to 1. Figure 4 and Supplementary Figures 1 and 2 suggest that under these scenarios it is not possible to estimate prevalence with any degree of accuracy.

Response:

Thank you for this comment which is clearly imperative for our methods to have general utility. It was also highlighted by reviewer 1 (please see point 5 above for detailed response and changes to manuscript). To summarise the above we have put a lot of focus into the revised manuscript around using more clinically relevant parameters as we agree this is key and we think this now greatly improves the relevance of our findings. We have shown key results figures (fig 3 and fig 5) with a minimum disease prevalence of 10% and using GRS AUC ranging upwards from 0.6. We have also added table 1 to show the minimum number of cases for each method to allow clinically meaningful estimate precision (CI +/- 5%) around a disease prevalence of 10%. We show prevalence lower than 10% can be estimated with more discriminative GRS or larger mixture size (sup figure 1) but in this scenario it might be worth considering if cases enrichment of the mixture cohort might be more appropriate. With regard to the heat maps (Sup Fig 1 and previous Sup Fig 2 now Figure 4) we have altered them to be clearer and emphasise that prevalence significantly lower than 0.5 can be estimated with accuracy.

We feel these changes to the manuscript highlight the generalisability of our approach even in situations with reduced genetic predisposition or rarer disease prevalence. We acknowledge there will be disease with minimal (AUC <0.6) polygenic risk where these approaches are probably not appropriate. Where disease prevalence is anticipated to be

rare (<10%) at a population level evaluation is possible by enriching for cases using careful choice of mixture criteria.

The coeliac disease example is an interesting use case of the proposed method. In an ascertained subsample, the prevalence of a rare disease might be high enough to get useful estimates. However, the genetic risk for coeliac disease is dominated by the DQ locus in the HLA region, which is probably the reason for the very unusual bimodal GRS distribution of the mixture population. This raises the question if there is any advantage in computing polygenic scores over simply tabulating the DQ genotypes in each of the groups and comparing the relative counts in order to estimate prevalence.

Response:

We are glad you thought coeliac disease an interesting use case and we feel highlights how our methods can address a traditionally difficult questions to answer. With regard to your question, unfortunately HLA genotyping data is rarely available in population cohorts. If imputing HLA from SNP data the additional effort of generating a polygenic risk score is minimal and previous work has shown additional discriminative information is provided from a polygenic risk score compared to just using the DQ locus alone [7]. We have further clarified this point in the text.

Added: to page 18 paragraph 2 starting line 397

In diseases where one variant has the predominant effect on genetic risk e.g., HLA-DQ in coeliac disease, it might be possible to estimate prevalence using just this variant. However previous work has shown a GRS including the predominant variant as well as smaller effect variants has better discriminative ability than the predominant variant alone [7].

While the assumption of a non-normal polygenic score distribution is the largest problem in my view, there are other issues which might affect all proposed methods, and which have not been investigated in the manuscript. For example,

how do population stratification and genetic heterogeneity between the GWAS cohort and the prediction cohorts affect the prevalence estimates?

Response:

We have tried to make this clearer throughout the manuscript emphasising the importance of the assumption of genetic equivalence between cases and non-cases and their respective reference cohorts. To allow our results to mirror the real world when using artificially generated data we have split the available case and control data into two to create reference cohorts containing different cases from the created mixture. Please see reviewer 1 point 1 for detail. We have also added details of steps to ensure equivalence in summary methods (Page 6 Paragraph 2) and cautions section (page 20 paragraph 3)

How does case ascertainment affect prevalence estimates? (Does it matter if the true prevalence is 1%, but only the most severe cases were recruited for the GWAS?)

Response:

If there is no genotype phenotype interaction it will have no impact. Where there is an interaction it will still not significantly affect prevalence estimates as long as that cohort is then not used for the reference case cohort. This is because if it is the reference case GRS will no longer mirror the mixture case GRS (unless identical recruitment criteria have been used). We have added clarification and an example of when this is an issue to the summary methods (Page 6 Paragraph 2) and cautions section (page 20 paragraph 3) in the discussion. Please also see reviewer 1 point 9 for more detailed discussion of this key point.

What is the effect of different genetic architectures?

Response:

This will manifest in the GRS distribution e.g. entirely polygenic being normally distributed versus a non-normally distributed distribution where small numbers of causal variants dominate. As discussed previously (point 1) our results are based on normally distributed GRS with all new methods working equally well. Non normal distributions may give minimal additional information when using KDE or EMD methods.

Investigating these issues may not be necessary if there was a clear demonstration that these methods allow accurate prevalence estimates regardless of potential issues like that. But in my opinion, the coeliac disease example falls short of this because the genetic score distribution seems to track one specific locus and is not representative of typical polygenic scores.

Response

We had generated an alternative example using the normally distributed polygenic type 2 diabetes GRS to estimate the proportion of type 2 diabetes cases in a population with proteinuria in the UK biobank and then compare this to the gold standard number of known type 2 diabetes cases (self-reported non-insulin treated diabetes cases or HbA1c \geq 48 to identify undiagnosed cases) with microalbuminuria. This shows a real world example of the accuracy of these methods around a prevalence of 13.4% using a polygenic GRS with AUC 0.65. We had previously not included this as we wanted to use a non-diabetes example. We have shown this below and would happily include at the editors discretion if felt to enhance the manuscript. We have included a summary of our methods which would be incorporated into the main manuscript accordingly.

Method: Calculating proportion of diabetes cases in individuals in UK biobank with microalbuminuria

Cases of microalbuminuria ($n = 18,697$) were taken from unrelated individuals of white European descent in the UK Biobank. We used albumin creatinine ratio (ACR) calculated from baseline assessment. In UKBB, a continuous measure of ACR was derived using urinary measures of albumin and creatinine. Microalbuminuria was defined based on international cut-offs ≥ 2.5 mg/mmol in males and ≥ 3.5 mg/mmol in females. As estimating cases of type 2 diabetes individuals analysis was restricted to exclude insulin treated diabetes cases. Diabetes was defined as self-reported or $\text{HbA1C} \geq 48$ mmol/mol to identify undiagnosed cases. Reference controls used were white European participants without diabetes and cases were non-insulin treated diabetes cases from UK Biobank without microalbuminuria. The true proportion of non-insulin treated type 2 diabetes cases was calculated by evaluating the number of white European non-insulin treated diabetes cases with microalbuminuria $n = 2509$.

Result:

Fig. Y illustrates a worked example using T2DGRS to estimate non-insulin treated diabetes cases within a cohort with microalbuminuria. All three new methodologies provide robust estimates of the proportion of individuals with type 2 diabetes with their 95% CIs (square brackets) encompassing the known proportion of 13.4%: Means = 14.8% [9.9%, 19.4%], EMD = 15.0% [10.4%, 19.2%], KDE = 17.8% [12.2%, 23.2%]. The 13.4% ground truth value was estimated using the reported number of type 2 diabetes cases within the microalbuminuria cohort in UK Biobank. Conversely the Excess method performs poorly (5.2% [0.2%, 3.9%]) and does not capture the known proportion.

Reviewer #3 (Remarks to the Author):

In this study, Evans et al compare different methods to estimate disease prevalence from the distribution of polygenic scores in a sample of homogeneous ancestry. I found the study not really clear in the presentation of its results and had to wait until the discussion to see somewhat clear conclusions. I think there is limited added value over the study from Thomas et al. (ref. 5) and therefore I don't see the novelty and why that study would

interest a wide audience. I do not recommend it for publication in Nature Communications.

Response:

Thank you for your comments we have considerably altered the manuscript to make the presentation of results and conclusions clearer throughout. With regard to limited added value over our previous study we politely disagree. This new manuscript clearly demonstrate that the Excess method works only in special cases (like using the T1DGRS with high AUC (0.88)) and is therefore not generalizable. This paper exposes the limitation of the Excess method whilst building on the concept of genetic stratification to describe three novel methods with considerable improvement in accuracy and precision. We explore the impact of key parameters on the accuracy of estimates providing a framework for researchers to use genetic stratification to answer novel questions of many polygenic diseases including those with lower genetic predisposition.

References

1. Kerminen, S., et al., *Geographic Variation and Bias in the Polygenic Scores of Complex Diseases and Traits in Finland*. Am J Hum Genet, 2019. **104**(6): p. 1169-1181.
2. Martin, A.R., et al., *Human Demographic History Impacts Genetic Risk Prediction across Diverse Populations*. Am J Hum Genet, 2017. **100**(4): p. 635-649.
3. Wellcome Trust Case Control, C., *Genome-wide association study of 14,000 cases of seven common diseases and 3,000 shared controls*. Nature, 2007. **447**(7145): p. 661-78.
4. Smith, G.D. and S. Ebrahim, 'Mendelian randomization': can genetic epidemiology contribute to understanding environmental determinants of disease? Int J Epidemiol, 2003. **32**(1): p. 1-22.
5. Davies, N.M., M.V. Holmes, and G. Davey Smith, *Reading Mendelian randomisation studies: a guide, glossary, and checklist for clinicians*. BMJ, 2018. **362**: p. k601.
6. Allen, N.E., et al., *UK biobank data: come and get it*. Sci Transl Med, 2014. **6**(224): p. 224ed4.
7. Sharp, S.A., et al., *A single nucleotide polymorphism genetic risk score to aid diagnosis of coeliac disease: a pilot study in clinical care*. Aliment Pharmacol Ther, 2020. **52**(7): p. 1165-1173.

Reviewers' Comments:

Reviewer #1:

Remarks to the Author:

The authors added a substantial part regarding the testing of the method under different parameters for sample size, case prevalence and moreover GRS prediction power, thus highlighting the areas of application and the minimal requirements for the method to work. The limitations and the potential of the method are now both carefully discussed. I therefore consider all my previous comments thoroughly answered. I have a couple of minor comments for this version of the manuscript, but mostly related to the form:

1. The new sections are very much needed and appreciated, but it seems the resulting manuscript here and there is a bit long and repetitive. I would suggest if and where possible to help the reader by condensing some sections and avoiding sentence repetitions in different parts of the manuscript, but that is ultimately an editorial matter so I leave that to the author's discretion.

2. Also figures might be reduced/condensed

2a. by avoiding to always show the reference and mixture distributions,

2b. by merging figure 1 and 2 in a single panel

3. In fig 4 caption T1D individuals are indicated as cases but elsewhere is said that a T2DGRS has been used in that figure, so cases shouldn't be T2D individuals here? I suspect that it is a typo, but if that is not the case please explain the rationale behind it.

4. In the section at p12 line 250 I find a bit confusing the reference first to figure 5 then to figure 4 ,where the latter has already been presented above, I suggest to invert the 2 sentences.

Reviewer #2:

Remarks to the Author:

The revised manuscript addresses many of the concerns that I and the other reviewers have raised. However, I still have the impression that the general utility of this approach is overstated, because it rests on assumptions that are usually not met in real data. I think there are specific cases where these methods can be useful in practice (prevalence estimates in subsets within a cohort rather than across cohorts), but in my opinion it would be important to limit the scope from the very beginning.

Supplementary Figure 1 demonstrates that it can generally not be assumed that genetic risk score distributions are equivalent among cases (or among non-cases) between different cohorts. The suggested solution to this problem is to select cases in a way which mirrors the selection criteria of the other cohort:

"BB WTCCC T2 is defined as first degree relative with type 1 diabetes diagnosed aged over 30 up to 40 years of age to recreate the WTCCC cohort"

This shifts the GRS distribution so that its mean is not significantly different from the GRS distribution in the Biobank data, but I don't think that an approach like this is robust enough that it can be relied on when case/control information is unavailable. Besides age of onset and disease status in relatives, there is potentially a large number of other variables that can affect the mean of the GRS distribution. These are generally not known and cannot be matched. This is the reason why genome wide association studies of dichotomous traits need to collect their own control samples, rather than re-use controls from previous studies.

Because of this, I don't think that prevalence estimates based on GRS are practical across different

cohorts. I do think that the methods discussed in this paper can be useful to estimate disease prevalence for the subset within a cohort for which case/control data is missing. In a case like this, it might be reasonable to assume that the GRS distribution is mostly influenced by disease prevalence, rather than by other factors.

The T2D/microalbuminuria example is interesting, and it is probably unaffected by across-cohort-heterogeneity, but it still suffers from the problem that it assumes that the GRS distribution in the mixture cohort (the microalbuminuria cases) is only affected by T2D prevalence. We know that this is not the case, because there is a high phenotypic and genetic correlation between T2D and microalbuminuria. I expect that this would shift the T2D GRS distribution even among microalbuminuria cases without T2D.

An experiment that would dispel this concern would be to take random subsets of the 18,697 microalbuminuria cases with T2D proportions varying from 0% to 100%, and show that these proportions are accurately recovered.

REVIEWER COMMENTS

Reviewer #1 (Remarks to the Author):

The authors added a substantial part regarding the testing of the method under different parameters for sample size, case prevalence and moreover GRS prediction power, thus highlighting the areas of application and the minimal requirements for the method to work. The limitations and the potential of the method are now both carefully discussed. I therefore consider all my previous comments thoroughly answered. I have a couple of minor comments for this version of the manuscript, but mostly related to the form: **Thank you for your review and we agree that the changes following the reviewers comments including the retesting and resultant changes have greatly improved our manuscript.**

1. The new sections are very much needed and appreciated, but it seems the resulting manuscript here and there is a bit long and repetitive. I would suggest if and where possible to help the reader by condensing some sections and avoiding sentence repetitions in different parts of the manuscript, but that is ultimately an editorial matter so I leave that to the author's discretion. **Thank you. On re-reading with this observation in mind we agree. We have identified sections (struck through) we feel (if the editors are in agreement) are repetitive and could be omitted/ significantly shortened and re-worded making it easier to read but without detriment to the clarity of the manuscript.**

2. Also figures might be reduced/condensed

2a. by avoiding to always show the reference and mixture distributions,: **This had been added to show how the GRS changed in different mixtures to give different estimates. We agree that they could be removed for figure 4 (now 3) and 6 (now 5) but feel it adds clarity in terms of the impact of proportion of cases to non-cases and mixture size in figure 3 (now 2). We are happy to remove from this figure if the editors would like us to.**

2b. by merging figure 1 and 2 in a single panel: **We agree this is a good idea and have done this**

3. In fig 4 caption T1D individuals are indicated as cases but elsewhere is said that a T2DGRS has been used in that figure, so cases shouldn't be T2D individuals here? I suspect that it is a typo, but if that is not the case please explain the rationale behind it. **Thank you for highlighting this it is indeed a typo and we have corrected it**

4. In the section at p12 line 250 I find a bit confusing the reference first to figure 5 then to figure 4 ,where the latter has already been presented above, I suggest to invert the 2 sentences. **We agree this is confusing. On reflection we don't think the sentences related to figure 4 adds to the manuscript and have deleted this.**

Reviewer #2 (Remarks to the Author):

The revised manuscript addresses many of the concerns that I and the other reviewers have raised. However, I still have the impression that the general utility of this approach is overstated, because it rests on assumptions that are usually not met in real data. I think there are specific cases where these methods can be useful in practice (prevalence estimates in subsets within a cohort rather than across cohorts), but in my opinion it would be important to limit the scope from the very beginning. Supplementary Figure 1 demonstrates that it can generally not be assumed that genetic risk score distributions are equivalent among cases (or among non-cases) between different cohorts. The suggested solution to this problem is to select cases in a way which mirrors the selection criteria of the other cohort: "BB WTCCC T2 is defined as first degree relative with type 1 diabetes diagnosed aged over 30 up to 40 years of age to recreate the WTCCC cohort"

This shifts the GRS distribution so that its mean is not significantly different from the GRS distribution in the Biobank data, but I don't think that an approach like this is robust enough that it can be relied on when case/control information is unavailable. Besides age of onset and disease status in relatives, there is potentially a large number of other variables that can affect the mean of the GRS distribution. These are generally not known and cannot be matched. This is the reason why genome wide association studies of dichotomous traits need to collect their own control samples, rather than re-use controls from previous studies. Because of this, I don't think that prevalence estimates based on GRS are practical across different cohorts. I do think that the methods discussed in this paper can be useful to estimate disease prevalence for the subset within a cohort for which case/control data is missing. In a case like this, it might be reasonable to assume that the GRS distribution is mostly influenced by disease prevalence, rather than by other factors.

Reply: Thank you for your re-review. As you highlight all these methods hinge on assumed genetic equivalence between reference and study cohorts. We fully agree with all your concerns and there will always remain a degree of uncertainty stemming from unknowns inherent to this sort of genetic analysis. We think changing the scope of the manuscript to predominantly focus on using these methods within a subset of a single dataset where references are derived from the same dataset such as UK Biobank is entirely reasonable as we agree that these methods can be very useful in this context. We have edited the manuscript to reflect this particularly the relevant sections of the methods outline and in the cautions section within the discussion (see below).

Whilst now shifting the focus of the manuscript to be predominantly about the application of the methods in subsets of a cohort we have still noted that the methods could be used with external reference cohorts as and when appropriate comparability conditions have been satisfied. In that regard we have worded more strongly the requirement for stringent pre-analysis checks that would only be possible in situations where clinical data for each cohort case/control data is available. We have also emphasised the hazards of this type of analysis and cautions around interpreting obtained results but do feel estimates made in

this context can be useful. This is highlighted by our original use of the methodology (Thomas et al Lancet Diabetes Endo 2018) giving comparable results to literature using non-genetic methods, (Thunander 2008 Diabetes Res Clin Pract). Unrecognised genetic factors can also impact clinical inferences made using other well established genetic techniques, for example Mendelian randomisation (Davies BMJ 2018). In Mendelian Randomisation, as with the approach described in this paper, sensitivity analyses, awareness of limitations and potential biases and cautious interpretation of results are essential.

Added to genetic stratification summary page 6

Finally, our methods are all based on the assumption that between the reference and mixture cohorts, cases and non-cases are genetically equivalent. This assumption must hold true for estimates to be valid and becomes less certain if the mixture cohort is derived from a different population than those used for reference. For this reason, we recommend these methods should be used to estimate disease prevalence within a subset of a population where reference cases and non-cases can be derived from the same population, for example UK Biobank. This does not completely exclude using reference cohorts derived from different datasets, particularly where robust disease cases may be difficult to define [7], but in this context extreme caution should be exercised prior to applying the methods and around the interpretation of the generated estimates. Using reference cohorts from a different population from the mixture analysis should only be undertaken following close examination of the selection criteria and demographics of the reference and mixture cohorts to ensure equivalence. This is of particular importance when studying different geographical populations where allele frequencies are known to vary [12, 13]. Accordingly, in this manuscript all analyses are restricted to white Europeans; the populations that the reference GRS distributions were derived from. Where possible, the GRS of the reference non-cases (controls) and cases should be compared with the GRS of known non-cases and cases within the same population the mixture has been taken from.

Added to cautions in discussion top of page 21

The use of genetic data in the context of genetic stratification means certain assumptions must hold true for the estimates to be valid. The same assumptions required for Mendelian randomisation [1, 17] should be met here. Key to the accuracy of estimates is the equivalence assumption which states that cases and non-cases in the mixture reflect their respective reference cohorts. The importance of meeting this assumption and the implications if it is not met, are highlighted by our example of a raised T2DGRS for an enriched reference T2D population from the WTCCC [8]. For this reason, to help ensure equivalence is maintained we recommend these methods are used in subsets of a cohort allowing reference cases and non-cases to be derived from the same dataset. If these methods are to be used with reference cohorts from different datasets, as was done previously [7], the equivalence assumption should be rigorously tested prior to analysis. This must initially involve detailed assessment of the selection criteria for the mixture and

reference cohorts and available literature, followed by comparison of the GRS between definite non-cases and cases from within the mixture and their respective references.

The T2D/microalbuminuria example is interesting, and it is probably unaffected by across-cohort-heterogeneity, but it still suffers from the problem that it assumes that the GRS distribution in the mixture cohort (the microalbuminuria cases) is only affected by T2D prevalence. We know that this is not the case, because there is a high phenotypic and genetic correlation between T2D and microalbuminuria. I expect that this would shift the T2D GRS distribution even among microalbuminuria cases without T2D.

An experiment that would dispel this concern would be to take random subsets of the 18,697 microalbuminuria cases with T2D proportions varying from 0% to 100%, and show that these proportions are accurately recovered.

Reply: Thank you for highlighting the potential impact of a genotype phenotype interaction in this example. We have performed the experiment as suggested (figure 1 below). This shows that at proportions of T2D of $\approx 30\%$ and under, precise estimates are recovered, covering the known 13.4% proportion of T2D which is correctly returned by all but the Excess method. At higher proportions of T2D the performance reduces so under-estimates are returned. This reflects the fact that in T2D cases there is a subtle reduction in mean T2DGRS in those with microalbuminuria (6.93 (SD 0.45)) compared to those without (6.98 (SD 0.46)). This result may reflect collider bias because the microalbuminuria phenotype reflects a cohort with higher multifactorial environmental risk for T2D so T2D occurs with less T2D genetic predisposition. There was no difference in mean T2DGRS in non-T2D cases with (6.77 (SD 0.46)) or without microalbuminuria (6.77 (SD 0.46)). This explains the excellent performance away from high T2D proportions. Whilst we feel this T2D ACR example nicely highlights the robust intrinsic performance of the methods using genetic risk scores despite low discrimination (AUC 0.65) it also further illustrates the hazards and therefore cautions of using genetics in this sort of context. Within the manuscript we have now further emphasised the importance of comparing the GRS of known mixture cases and controls with their respective reference counterparts to try and avoid this issue. With this in mind despite the accurate returned estimates the ACR example is clearly not the best one to use to show that the methods can perform with low AUC GRS. We have therefore generated an alternative example that could be included in the manuscript along the same lines but identifying type 2 diabetes within participants with glaucoma. This example still shows the real world performance of the methods using a polygenic GRS with AUC 0.65 but is unaffected by such interactions and returns accurate estimates with T2D proportions for 0 to 100% (figure 2 below). This allows accurate estimate of the number of type 2 diabetes cases within a population with glaucoma (type 2 diabetes being a risk factor) to be returned (figure 3).

Added in cautions

Careful GRS comparison between the reference cohorts and definite cases and non-cases from the mixture will also help mitigate any potential impact of unrecognized genotype phenotype interactions which may arise when selecting subgroups. Furthermore we

recommend careful investigation for overlapping genetic associations and pleiotropy using standard Mendelian Randomisation approaches (17, 18).

Figure 1: Estimates returned with known proportions of T2D within a population with microalbuminuria

Figure 2: Estimates returned with known proportions of T2D within a population with glaucoma

Figure 3: Performance of methods in identifying type 2 diabetes cases in a mixture cohort of self reported glaucoma cases.

Fig. 3 illustrates a worked example using T2DGRS to estimate non-insulin treated diabetes cases within a cohort with glaucoma. All three new methodologies provide robust estimates of the proportion of individuals with type 2 diabetes with their 95% CIs (square brackets) encompassing the known proportion of 6.6%: Means = 7.9% [1.9%, 14.4%], EMD = 8.2% [2.4%, 14.9%], KDE = 8.4% [0.3%, 16.0%]. The 6.6% ground truth value was estimated using the reported number of type 2 diabetes cases within the glaucoma cohort in UK Biobank. Conversely the Excess method performs poorly (2.1% [0.8%, 5.8%]) and does not capture the known proportion. Cases of self-reported glaucoma ($n=9857$) were taken from unrelated individuals of white European descent in the UK Biobank. As estimating cases of type 2 diabetes analysis was restricted to exclude insulin treated diabetes cases. Diabetes was defined as self-reported or $HbA1C \geq 48$ mmol/mol to identify undiagnosed cases. Reference controls used were white European participants without diabetes and glaucoma and cases were non-insulin treated diabetes cases from UK Biobank without glaucoma.

Reviewers' Comments:

Reviewer #1:

Remarks to the Author:

The Authors replied to all the minor concerns raised in my previous revision.

Personally, I feel that after the corrections the title should be adjusted to match the scope of the main text, avoiding the word "population" which sounds too general, perhaps substituting with "cohort" or similar.

Reviewer #2:

Remarks to the Author:

The authors have carefully addressed all concerns which I have raised previously. Figures 1-3 in the rebuttal are clear and informative, and I think it could be a good idea to include these examples (T2D and microalbuminuria/glaucoma) in the supplementary material.

Reviewer #1 (Remarks to the Author):

The Authors replied to all the minor concerns raised in my previous revision.

Personally, I feel that after the corrections the title should be adjusted to match the scope of the main text, avoiding the word "population" which sounds too general, perhaps substituting with "cohort" or similar.

Thankyou we have changed this so title is now: **Estimating disease prevalence in large datasets using genetic risk scores**

Reviewer #2 (Remarks to the Author):

The authors have carefully addressed all concerns which I have raised previously. Figures 1-3 in the rebuttal are clear and informative, and I think it could be a good idea to include these examples (T2D and microalbuminuria/glaucoma) in the supplementary material.

Thankyou. We have included these as suggested